# One Sample Fits All: Approximating All Probabilistic Values Simultaneously and Efficiently

**Weida Li**
School of Computing
National University of Singapore
`vidaslee@gmail.com`

**Yaoliang Yu**
School of Computer Science
University of Waterloo
Vector Institute
`yaoliang.yu@uwaterloo.ca`

## Abstract

The concept of probabilistic values, such as Beta Shapley values and weighted Banzhaf values, has gained recent attention in applications like feature attribution and data valuation. However, exact computation of these values is often exponentially expensive, necessitating approximation techniques. Prior research has shown that the choice of probabilistic values significantly impacts downstream performance, with no universally superior option. Consequently, one may have to approximate multiple candidates and select the best-performing one. Although there have been many efforts to develop efficient estimators, none are intended to approximate all probabilistic values both simultaneously and efficiently. In this work, we embark on the first exploration of achieving this goal. Adhering to the principle of maximum sample reuse and avoiding amplifying factors, we propose a one-sample-fits-all framework parameterized by a sampling vector to approximate intermediate terms that can be converted to any probabilistic value. Leveraging the concept of $(\epsilon, \delta)$-approximation, we theoretically identify a key formula that effectively determines the convergence rate of our framework. By optimizing the sampling vector using this formula, we obtain i) a one-for-all estimator that achieves the currently best time complexity for all probabilistic values on average, and ii) a faster generic estimator with the sampling vector optimally tuned for each probabilistic value. Particularly, our one-for-all estimator achieves the fastest convergence rate on Beta Shapley values, including the well-known Shapley value, both theoretically and empirically. Finally, we establish a connection between probabilistic values and the least square regression used in (regularized) datamodels, showing that our one-for-all estimator can solve a family of datamodels simultaneously. Our code is available at `https://github.com/watml/one-for-all`.

## 1 Introduction

The problem of attribution is central in many aspects of machine learning (Rozemberczki et al. 2022). Examples include data valuation (Ghorbani and Zou 2019), feature attribution (Lundberg and Lee 2017), multi-agent reinforcement learning (Wang et al. 2022), data attribution (Ilyas et al. 2022), and the list goes on. One popular methodology is to leverage the concept of probabilistic values, which is uniquely characterized by the axioms of linearity, null, monotonicity and symmetry in cooperative game theory (Weber 1988). Recent studies demonstrate that downstream performance employing this concept relies on the choice of probabilistic values, and the best one varies (Kwon and Zou 2022b; Li and Yu 2023). Therefore, practitioners may resort to approximating multiple candidates of probabilistic values and then select the best-performing one (Kwon and Zou 2022b).

38th Conference on Neural Information Processing Systems (NeurIPS 2024).

In general, probabilistic values can only be approximated as they require exponentially many utility evaluations to compute exactly. Thus far, there has been a line of works devoted to designing efficient estimators for the Shapley value (e.g., Covert and Lee 2021; Jia et al. 2019; Kolpaczki et al. 2024; Zhang et al. 2023), while Li and Yu (2023) and Wang and Jia (2023b) proposed efficient estimators specific to weighted Banzhaf values. Despite recent progress in research on generic estimators for approximating any probabilistic value (Li and Yu 2024; Lin et al. 2022), none of them can approximate all probabilistic values *simultaneously and efficiently*. Overall, there is a strong demand for efficient one-for-all estimators, the possibilities of which will be explored in this work.

To sum up, we propose a **O**ne-sample-**F**its-**A**ll (OFA) framework parameterized by a sampling vector to approximate intermediate terms that can be converted to any probabilistic value. Particularly, our framework i) adheres to the principle of maximum sample reuse and ii) does not include amplifying factors in the conversion. These two properties are considered indispensable as we observe that i) the empirical fastest estimators designed for the Shapley value or weighted Banzhaf values all follow she principle of maximum sample reuse and ii) amplifying factors could deteriorate the convergence rates of estimators. Then, using the concept of $(\epsilon, \delta)$-approximation, i.e., $P(\|\hat{\phi} - \phi\|_2 \geq \epsilon) \leq \delta$ where $\phi$ refers to some probabilistic value and $\hat{\phi}$ is its estimate, we theoretically identify a formula from our framework that effectively determines the corresponding convergence rate, through which the sampling vector can be optimized. Specifically, we deduce i) an efficient one-for-all estimator (OFA-A) while optimizing the formula for all probabilistic values on **A**verage and ii) a faster generic estimator (OFA-S) while the optimization is done for each **S**pecific probabilistic value. The results of our convergence analysis are summarized as follows:

(i) Our OFA-A achieves the convergence rate $O(n \log n)$ for all probabilistic values on average. Notably, $O(n \log n)$ is the currently-known best time complexity for *some* probabilistic values.

(ii) For Beta Shapley values parameterized by $\alpha, \beta \geq 1$ (Kwon and Zou 2022a), our OFA-A estimator requires $O(n \log n)$ utility evaluations to achieve an $(\epsilon, \delta)$-approximation *simultaneously*. Note that $\alpha = \beta = 1$ corresponds to the commonly-used Shapley value (Shapley 1953). For the Shapley value, the previous best convergence rate is $O(n(\log n)^2)$,[1] achieved by the group testing estimator (Wang and Jia 2023a, Theorem 6); however, we note that in our experiments the previous best-performing estimator is the complement estimator (Zhang et al. 2023), whose convergence rate is unknown. For Beta Shapley values with $(\alpha = 1, \beta > 1)$ or $(\alpha > 1, \beta = 1)$, the previous best estimator requires $O(n(\log n)^3)$ utility evaluations instead (Li and Yu 2024, Proposition 4 and Remark 3).

(iii) For weighted Banzhaf values parameterized by $0 < w < 1$, the time complexity of our OFA-A is $O(n^{\frac{3}{2}} \log n)$, not rivaling the previous best convergence rate $O(n \log n)$ achieved by the estimator exclusive to weighted Banzhaf values (Li and Yu 2023, Proposition 2). Nevertheless, our OFA-S achieves the convergence rate of $O(n \log n)$ for both Beta Shapley values (with $\alpha, \beta \geq 1$) and weighted Banzhaf values.

In our experiments, the empirical convergence rates align well with the theoretical ones derived using the concept of $(\epsilon, \delta)$-approximation. Additionally, we establish a connection between probabilistic values and the least square regressions employed in datamodels (Ilyas et al. 2022), demonstrating that our OFA-A estimator can solve a family of datamodels simultaneously if it is the distances between feature coordinates that matter. This condition is met while using datamodels to detect similar training examples to a given target. Furthermore, we also identify a group of regularized datamodels that our OFA-A estimator can solve simultaneously without this condition.

## 2 Preliminaries

Let $n$ be the number of players and $[n] := \{1, 2, \ldots, n\}$ be the set of all players. In data valuation (feature attribution, respectively), $n$ refers to the number of training data (features, respectively). For simplicity, we write $S \backslash i$ and $S \cup i$ instead of $S \backslash \{i\}$ and $S \cup \{i\}$, respectively. Meanwhile, (lowercase) $s$ denotes the cardinality of the set (uppercase) $S$. Then, each probabilistic value can be written as

$$\phi_i = \phi_i(U) = \sum_{S \subseteq [n] \backslash i} p_{s+1}[U(S \cup i) - U(S)]$$

---

[1]Using another definition, Musco and Witter (2024) claimed to have an estimator with convergence rate $O(n \log n)$ for the Shapley value after this work was accepted.

where $U : 2^{[n]} \to \mathbb{R}$ is a utility function and $\mathbf{p} \in \mathbb{R}^n$ is a non-negative vector such that $\sum_{s=1}^{n} \binom{n-1}{s-1} p_s = 1$. Take data valuation as an example, $U(S)$ may measure the performance of models trained on $S \subseteq [n]$, with which $\phi_i(U)$ can be interpreted as the contribution of the $i$-th data point to the performance of models trained on $[n]$.

If there exists a (Borel) probability measure $\mu$ on the closed interval $[0, 1]$ such that $p_s = \int_0^1 w^{s-1}(1-w)^{n-s} \mathrm{d}\mu(w)$, then the resulting probabilistic value is referred to as a semi-value (Dubey et al. 1981). If $\mu$ represents a Dirac delta distribution $\delta_a$, the corresponding probabilistic value is referred to as the weighted Banzhaf value parameterized by $a$, or WB-$a$. For Beta Shapley values, denoted by Beta$(\alpha, \beta)$, $\mu(A) = \int_A w^{\beta-1}(1-w)^{\alpha-1} \mathrm{d}w$. In practice, the considered range of $\alpha$ or $\beta$ is $[1, \infty)$ (Kwon and Zou 2022a,b). Particularly, Beta$(1, 1)$, whose $\mu$ is the uniform distribution (over $[0, 1]$), corresponds to the Shapley value.

We will use the standard notion of $(\epsilon, \delta)$-approximation to analyze a (randomized) estimate $\hat{\phi}$ of some probabilistic value $\phi$.

**Definition 1.** *We say a (randomized) estimate $\hat{\phi}$ achieves an $(\epsilon, \delta)$-approximation of $\phi$ if $P(\|\hat{\phi} - \phi\|_2 \geq \epsilon) \leq \delta$.*

For instance, Wang and Jia (2023b, Theorem 4.9) proved that their proposed estimator requires $O(\frac{n}{\epsilon^2} \log \frac{n}{\delta})$ utility evaluations to achieve an $(\epsilon, \delta)$-approximation for WB-0.5, provided that $\|U\|_\infty \leq 1$. When $\epsilon$ and $\delta$ are considered fixed constants, we then simply say the estimator converges at $O(n \log n)$ rate.

# 3   Motivations

**One-For-All Estimators**   In this paper, an estimator is referred to as one-for-all if it is capable of sampling subsets **O**nce to approximate **A**ll probabilistic values.

Though existing estimators are not designed to approximate all probabilistic values simultaneously, some of them can be easily modified for this end by using the weighted sampling technique. Take the sampling lift (SL) estimator (Moehle et al. 2022) as an example, its approximation is based on

$$\phi_i = \mathbb{E}_{S \subseteq [n] \setminus i}[U(S \cup i) - U(S)] \text{ where } P(S) = p_{s+1}.$$

If we fix the probability of sampling $S$ to be the one, denoted by $\mathbf{q} \in \mathbb{R}^n$, for the Shapley value, there is

$$\phi_i = \mathbb{E}_{S \subseteq [n] \setminus i}^{\text{Shap}} \left[ \frac{p_{s+1}}{q_{s+1}} (U(S \cup i) - U(S)) \right], \tag{1}$$

which is the weighted sampling lift (WSL) estimator employed by Kwon and Zou (2022a). Therefore, we can store the accumulated results $\{U(S \cup i) - U(S)\}$ separately for each subset size of $S$ so that they can be reweighted to be any probabilistic value.

**The Effect of Amplifying Factors**   However, the scalars $\{\frac{p_{s+1}}{q_{s+1}}\}$ potentially introduce a non-negligible factor into the theoretical convergence rate. To demonstrate, we take the WSL estimator as an example. In this case, $\hat{\phi}_i = \frac{1}{T} \sum_{t=1}^{T} X_t$ where $\{X_t\}_{t=1}^{T}$ are i.i.d. random variable such that $P(X_t = \frac{p_{s+1}}{q_{s+1}}(U(S \cup i) - U(S))) = q_{s+1}$ and thus $\mathbb{E}[X_t] = \phi_i$. Assume that $\|U\|_\infty \leq 1$, by the Hoeffding's inequality, $P(|\hat{\phi}_i - \phi_i| \geq \epsilon) \leq 2 \exp\left(-\frac{T\epsilon^2}{8C^2}\right)$ where $C = \max_{1 \leq k \leq n} \frac{p_k}{q_k}$. By solving $2 \exp\left(-\frac{T\epsilon^2}{8C^2}\right) \leq \delta$, we eventually obtain $T \geq \frac{8C^2}{\epsilon^2} \log \frac{2}{\delta}$ and therefore the convergence rate of $\hat{\phi}_i$ is $O(\frac{C^2}{\epsilon^2} \log \frac{2}{\delta})$. Consequently, if $C \to \infty$ as $n \to \infty$, this theoretical convergence rate deteriorates asymptotically. For the Banzhaf value, $p_k = \frac{1}{2^{n-1}}$; since $q_k = \frac{(k-1)!(n-k)!}{n!}$, if $k = \frac{n+1}{2}$, there is $\frac{p_k}{q_k} \in \Theta(n^{\frac{1}{2}})$ by the Stirling's approximation $d! \simeq \sqrt{d} \left(\frac{d}{e}\right)^d$. Therefore, $C^2$ introduces a factor of $\Theta(n)$ into the theoretical convergence rate, though the derived formula may not be tight. If we switch the roles of $\mathbf{p}$ and $\mathbf{q}$, the introduced factor $C^2$ is as worst as $\Theta(2^{2n})$. To generalize this idea, a formula is said to contain an amplifying factor if it involves $\gamma \cdot U(S)$ such that $\gamma \to \infty$ as $n \to \infty$.

Regarding this, we notice that Kwon and Zou (2022b) resort to a one-for-all estimator based on

$$\phi_i = \sum_{s=1}^{n} m_s \cdot \mathbb{E}_{\substack{R \subseteq [n] \setminus i \\ r = s-1}}[U(R \cup i) - U(R)] \tag{2}$$

where $m_s = \binom{n-1}{s-1} p_s$ and each expectation is taken over the corresponding uniform distribution. We refer to this estimator as weightedSHAP in this work. As can be verified, Eq. (2) does not contain any amplifying factors, i.e., each $m_s$ does not grow as $n \to \infty$.

**The Principle of Maximum Sample Reuse** However, estimators designed according to Eqs. (1) and (2) are not expected to be efficient as it does not obey the principle of maximum sample reuse. Precisely, an estimator adheres to the principle of maximum sample reuse if each sampled subset is used to update all estimates $\{\hat{\phi}_i\}_{i \in [n]}$. As analyzed by Zhang et al. (2023, Section 4.2), estimators based on sampled marginal contributions $\{U(S \cup i) - U(S)\}$ are impossible to meet the principle of maximum sample reuse. By contrast, we observe that the SHAP-IQ estimator proposed by Fumagalli et al. (2024) can also be adopted for this end, which employs the formula

$$\phi_i = p_n(U([n]) - U(\emptyset)) + 2H \mathbb{E}_{\emptyset \subsetneq S \subsetneq [n]}[((n-s)m_s \mathbb{1}_{i \in S} - s m_{s+1} \mathbb{1}_{i \notin S})(U(S) - U(\emptyset))] \tag{3}$$

where $m_s = \binom{n-1}{s-1} p_s$, $H = \sum_{j=1}^{n-1} \frac{1}{j}$, and $P(S) \propto \binom{n-2}{s-1}^{-1}$. In particular, SHAP-IQ is equal to the unbiased KernelSHAP estimator (Covert and Lee 2021) for the Shapley value; see (Fumagalli et al. 2024, Theorem 4.5). Although SHAP-IQ follows the principle of maximum sample reuse, it is apparent that Eq. (3) contains an amplifying factor $H \in \Theta(\log n)$. Meanwhile, there is another line of research in quest of efficient estimators for the Shapley value by reducing the variance via the stratified sampling technique (Burgess and Chapman 2021; Castro et al. 2017; Maleki et al. 2013; Wu et al. 2023). However, such a technique also does not verify the principle of maximum sample reuse.

**Empirical Evidence** For convenience, we formally define the two aforementioned desirable properties for estimators to possess as **P1:** The underlying formula contains no amplifying factors and **P2:** Each sampled subset is used to update all the estimates $\{\hat{\phi}_i\}_{i=1}^{n}$. In Figure 1, we provide some experiment results while setting $n = 24$ to support our aforementioned informal analysis. Precisely, we implement six one-for-all estimators by combining the weighted sampling technique and the previous estimators. Some of our observations are:

(i) On WB-0.5, weightedSHAP, which satisfies **P1** but not **P2**, is empirically not comparable to MSR-Banzhaf that possesses both **P1** and **P2**. This observation supports the necessity of **P2**.

(ii) On WB-0.5, SHAP-IQ sticks to **P2** but not **P1**. It is clear that SHAP-IQ also performs significantly worse than MSR-Banzhaf, which highlights the role of **P1**.

(iii) The sudden rises of relative differences stem from the existence of significantly large amplifying factors. For WSL-Banzhaf on Beta(1, 1), the scalar for $U(i) - U(\emptyset)$ is as large as $\frac{2^n}{n}$!

In Table 1, we summarize the previous estimators in terms of **P1** and **P2**, and defer the technical details to Appendix D. Notably, the complement estimator is empirically the best for the Shapley value, while it is MSR for weighted Banzhaf values; both of them follow **P1** and **P2**.

Table 1: A scope of "all" indicates that the estimator can approximate any probabilistic value, whereas "weighted Banzhaf" suggests that the estimator can only approximate weighted Banzhaf values. **P1** refers to the property that the underlying formula does not contain any amplifying factors *for all probabilistic values in its scope*, while **P2** means whether each sampled subset is used to update all the estimates $\{\hat{\phi}_i\}_{i=1}^{n}$. For AME, the range of $\gamma$ in $\gamma \cdot U(S)$ could be $(0, \infty)$, independent of $n$. Originally, AME only applies to a subfamily of semi-values, but we extend it for all semi-values in Appendix D.

| | WSL (Kwon and Zou 2022a) | SL (Moehle et al. 2022) | GELS (Li and Yu 2024) | ARM (Kolpaczki et al. 2024) | MSR (Wang and Jia 2023b) | SHAP-IQ (Fumagalli et al. 2024) | weightedSHAP (Kwon and Zou 2022b) |
|---|---|---|---|---|---|---|---|
| scope | all | Shapley | all | all | weighted Banzhaf | all | all |
| **P1** | ✗ | ✓ | ✗ | ✓ | ✓ | ✗ | ✓ |
| **P2** | ✗ | ✗ | ✗ | ✗ | ✓ | ✓ | ✗ |

| | permutation (Castro et al. 2009) | kernelSHAP (Lundberg and Lee 2017) | unbiased kernelSHAP (Covert and Lee 2021) | group testing (Wang and Jia 2023a) | complement (Zhang et al. 2023) | AME (Lin et al. 2022) | OFA (ours) |
|---|---|---|---|---|---|---|---|
| scope | Shapley | Shapley | Shapley | Shapley | Shapley | semi-values | all |
| **P1** | ✓ | ✗ | ✗ | ✗ | ✓ | ✗ | ✓ |
| **P2** | ✗ | ✓ | ✓ | ✗ | ✓ | ✓ | ✓ |

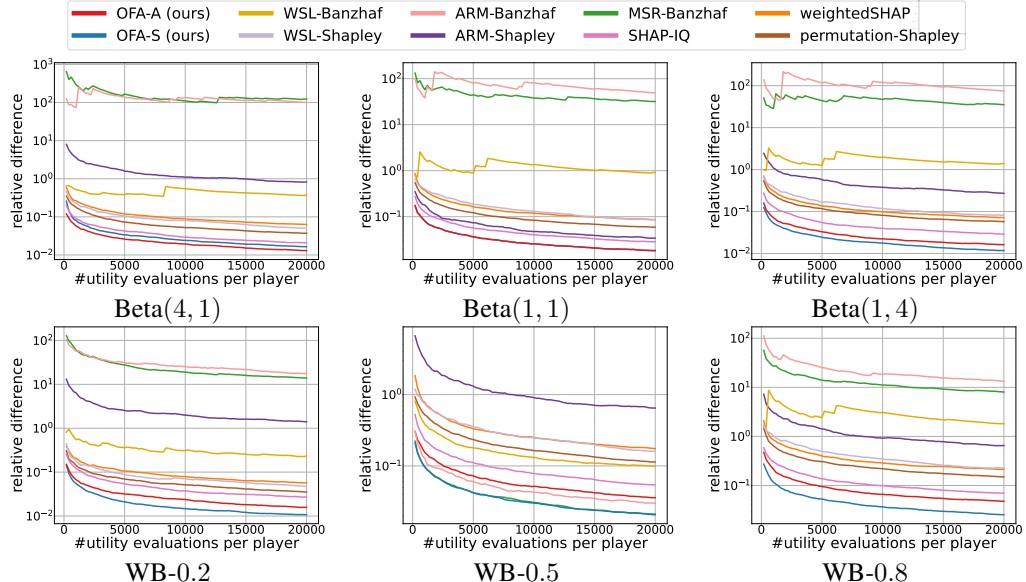

Figure 1: Comparison of ten one-for-all estimators. Beta$(\alpha, \beta)$ denotes Beta Shapley values, whereas WB-$a$ refers to weighted Banzhaf values. Our OFA-S estimator is equal to the OFA-A estimator for the Shapley value. The suffix "Shapley" indicates that there is no reweighting for the Shapley value, while "Banzhaf" stands for the Banzhaf value. The permutation estimator is originally proposed for the Shapley value. The utility function $U$ is the cross-entropy loss of LeNet trained on $24$ data from FMNIST. All the results are averaged using $30$ random seeds.

## 4 Main Results

The framework we propose is built upon

$$\phi_i = \sum_{s=1}^{n} m_s \cdot \left( \mathop{\mathbb{E}}_{\substack{i \in R \\ r=s}}[U(R)] - \mathop{\mathbb{E}}_{\substack{i \notin R \\ r=s-1}}[U(R)] \right) \tag{4}$$

where $m_s = \binom{n-1}{s-1}p_s$ and each expectation is taken over the corresponding uniform distribution. For simplicity, we write $\phi_{i,s}^+ = \mathbb{E}_{i \in R, r=s}[U(R)]$ and $\phi_{i,s-1}^- = \mathbb{E}_{i \notin R, r=s-1}[U(R)]$. Clearly, there is no amplifying factors in Eq. (4). Meanwhile, the structure of Eq. (4) suits the principle of maximum sample reuse. Since $\{\phi_{i,k}^+\}_{k=1,n-1,n}$ and $\{\phi_{i,k}^-\}_{k=0,1,n-1}$ can be calculated exactly using $2n+2$ utility evaluations of $U$, our focus is to efficiently approximate $\{\phi_{i,s}^+, \phi_{i,s}^-\}_{2 \leq s \leq n-2}$. The proposed framework is presented in Algorithm 1; $q_j$ is the probability of drawing a subset of $[n]$ with size $j+1$. To facilitate the choice of the sampling vector $\mathbf{q} \in \mathbb{R}^{n-3}$ appearing in Algorithm 1, our first step is to theoretically ascertain a key formula that effectively determines the convergence rate of Algorithm 1.

**Theorem 1.** *Assume i)* $\|U\|_\infty \leq u$ *and ii)* $0 < \epsilon \leq \sqrt{2D(\mathbf{m}, \mathbf{q})\gamma(\mathbf{q})^2 u^2}$. *For* $\hat{\phi}$ *in Algorithm 1, it requires* $\frac{4nu^2 D(\mathbf{m}, \mathbf{q})}{\epsilon^2} \log \frac{8n^2}{\delta}$ *evaluations of* $U$ *to achieve* $P(\|\hat{\phi} - \phi\|_2 \geq \epsilon) \leq \delta$ *where*

$$D(\mathbf{m}, \mathbf{q}) = \sum_{s=2}^{n-2} \frac{n}{q_{s-1}} \left( \frac{m_s^2}{s} + \frac{m_{s+1}^2}{n-s} \right) \text{ and } \gamma(\mathbf{q}) = \min_{2 \leq s \leq n-2} \min \left( \frac{q_{s-1} \cdot s}{n}, \frac{q_{s-1} \cdot (n-s)}{n} \right).$$

We remark that $D(\mathbf{m}, \mathbf{q})$ is jointly convex in $\mathbf{m}$ and $\mathbf{q}$. The second assumption in Theorem 1 can be removed if we pre-allocate the number of sampled subsets for each $\phi_{i,s}^+$ or $\phi_{i,s}^-$ and draw subsets in a predetermined order; see the proof in Appendix A for details. Precisely, let $T_{i,s}^+$ be the number of subsets for estimating $\phi_{i,s}^+$, and define $T_{i,s}^-$ similarly; then the pre-allocated numbers are $T_{i,s}^+ \approx \frac{s \cdot q_{s-1}}{n}T$ and $T_{i,s}^- \approx \frac{(n-s)q_{s-1}}{n}T$, which are the expected values of $T_{i,s}^+$ and $T_{i,s}^+$ while using Algorithm 1; $T$ refers to the total number of sampled subsets. By Theorem 1, the convergence rate of Algorithm 1 is $O(D(\mathbf{m}, \mathbf{q}) \cdot n \log n)$, and thus achieving the currently best convergence rate $O(n \log n)$ requires $D(\mathbf{m}, \mathbf{q}) \in O(1)$.

**Algorithm 1:** The **O**ne-**S**ample-**F**its-**A**ll (OFA) Framework

**Input:** A utility function $U : 2^{[n]} \to \mathbb{R}$, a positive probability vector $\mathbf{q} \in \mathbb{R}^{n-3}$, and a total number $T$ of samples

**Output:** Estimates to $\phi_{i,k^+}^+$ and $\phi_{i,k^-}^-$ with $i, k^+ \in [n]$ and $0 \leq k^- \leq n-1$

1   $\hat{\phi}_{i,k^+}^+ \leftarrow \phi_{i,k^+}^+$ and $\hat{\phi}_{i,k^-}^- \leftarrow \phi_{i,k^-}^-$ for $i \in [n]$, $k^+ \in \{1, n-1, n\}$ and $k^- \in \{0, 1, n-1\}$

2   $\hat{\phi}_{i,k}^+ \leftarrow 0, T_{i,k}^+ \leftarrow 0, \hat{\phi}_{i,k}^- \leftarrow 0$ and $T_{i,k}^- \leftarrow 0$ with $i \in [n]$ and $2 \leq k \leq n-2$

3   **for** $t = 1, 2, \ldots, T$ **do**

4      Sample $s_t$ from $\{2, 3, \ldots, n-2\}$ according to $\mathbf{q}$

5      Sample $S_t$ uniformly from $\{R \subseteq [n] \mid r = s_t\}$

6      $v \leftarrow U(S_t)$

7      **for** $i = 1, 2, \ldots, n$ **do**

8         **if** $i \in S_t$ **then**

9            $\hat{\phi}_{i,s_t}^+ \leftarrow \frac{T_{i,s_t}^+}{T_{i,s_t}^+ + 1} \hat{\phi}_{i,s_t}^+ + \frac{1}{T_{i,s_t}^+ + 1} v$ and $T_{i,s_t}^+ \leftarrow T_{i,s_t}^+ + 1$

10        **else**

11            $\hat{\phi}_{i,s_t}^- \leftarrow \frac{T_{i,s_t}^-}{T_{i,s_t}^- + 1} \hat{\phi}_{i,s_t}^- + \frac{1}{T_{i,s_t}^- + 1} v$ and $T_{i,s_t}^- \leftarrow T_{i,s_t}^- + 1$

12   **Aggregation Phase:** $\hat{\phi}_i = \sum_{s=1}^n m_s (\hat{\phi}_{i,s}^+ - \hat{\phi}_{i,s-1}^-)$

## 4.1   A One-For-All Estimator

To obtain our one-for-all estimator, our goal is to find a $\mathbf{q}^{\text{OFA-A}}$ such that $D(\mathbf{m}, \mathbf{q}^{\text{OFA-A}}) \in O(1)$ for as many $\mathbf{m}$ as possible. To this end, we define $\mathbf{q}^{\text{OFA-A}}$ to be the uniquely optimal solution to

$$\underset{\mathbf{q} \in \mathbb{R}^{n-3}}{\operatorname{argmin}} \overline{D}(\mathbf{q}) = \int_{\mathbf{m} \in \Delta} D(\mathbf{m}, \mathbf{q}) \mathrm{d}\nu(\mathbf{m})$$

where $\Delta = \{\mathbf{m} \in \mathbb{R}^n \mid m_s \geq 0 \text{ and } \sum_{s=1}^n m_s = 1\}$ and $\nu$ is the uniform distribution on $\Delta$. In our work, our OFA-A estimator refers to the use of $\mathbf{q}^{\text{OFA-A}}$ in Algorithm 1.

**Proposition 1.** $\mathbf{q}_{s-1}^{OFA\text{-}A} \propto \frac{1}{\sqrt{(s)(n-s)}}$ *and* $\overline{D}(\mathbf{q}^{OFA\text{-}A}) \in O(1)$. *In other words, our OFA-A estimator achieves the convergence rate of* $O(n \log n)$ *simultaneously for all probabilistic values on average.*

Our next proposition provides a condition on $\mu$ for semi-values such that our OFA-A estimator achieves the convergence rate of $O(n \log n)$. In other words, we explicitly identify a subfamily of semi-values for which our OFA-A estimator achieves the currently best time complexity simultaneously.

**Proposition 2.** *Our OFA-A estimator achieves the convergence rate of* $O(n \log n)$ *simultaneously for all semi-values whose probability density functions exist and are bounded. Particularly, Beta Shapley values with* $\alpha, \beta \geq 1$ *all satisfy this condition.*

To our knowledge, the previous theoretically-fastest estimator for the Shapley value is demonstrated by Wang and Jia (2023a, Theorem 6) as $O(n(\log n)^2)$. By contrast, our OFA-A estimator achieves the convergence rate of $O(n \log n)$. Meanwhile, it also surpasses the previous best time complexity for Beta Shapley values with $(\alpha = 1, \beta > 1)$ or $(\alpha > 1, \beta = 1)$, which is $O(n(\log n)^3)$ (Li and Yu 2024, Proposition 4 and Remark 3). Remarkably, our OFA-A estimator enjoys this fastest convergence rate *simultaneously for a broad subfamily of probabilistic values* .

**Proposition 3.** *If* $p_s = a^{s-1}(1-a)^{n-s}$ *with* $0 < a < 1$, *which corresponds to the weighted Banzhaf value parameterized by* $w$, *then* $D(\mathbf{m}, \mathbf{q}^{OFA\text{-}A}) \in O(n^{\frac{1}{2}})$. *In other words, the OFA estimator achieves the convergence rate of* $O(n^{\frac{3}{2}} \log n)$ *simultaneously for all WB-a with* $0 < a < 1$.

The previous best convergence rate for weighted Banzhaf values is $O(n \log n)$ (Li and Yu 2023, Proposition 2), ours is slower by a factor of $n^{\frac{1}{2}}$. Nevertheless, we will demonstrate that our generic estimator, which is expected to be faster than our OFA-A estimator, achieves the best convergence rate for all weighted Banzhaf values.

## 4.2 A Faster Generic Estimator

Our faster generic estimator (OFA-S) is obtained via optimizing $\mathbf{q}$ for each **S**pecific $\mathbf{m}$. Precisely, for each $\mathbf{m}$, we have

$$\mathbf{q}_{s-1}^{\text{OFA-S}} \propto \sqrt{\frac{m_s^2}{s} + \frac{m_{s+1}^2}{n-s}} \quad \text{where} \quad \mathbf{q}^{\text{OFA-S}} = \operatorname*{argmin}_{\mathbf{q}\in\mathbb{R}^{n-3}} D(\mathbf{m},\mathbf{q}) \ \text{s.t.} \ \sum_{j=1}^{n-3} q_j = 1, \qquad (5)$$

which can be obtained using the Cauchy-Schwarz inequality. For the Shapley value, $\mathbf{q}^{\text{OFA-S}} = \mathbf{q}^{\text{OFA-A}}$. Our next proposition specifies a sufficient condition for semi-values such that $D(\mathbf{m}, \mathbf{q}^{\text{OFA-S}}) \in O(1)$.

**Proposition 4.** *For semi-values, $D(\mathbf{m}, \mathbf{q}^{OFA\text{-}S}) \in O(1)$ if i) $\mu$ has a bounded probability density function or ii) $\int_{(0,1)} \frac{1}{w(1-w)}\mathrm{d}\mu(w) < \infty$. Particularly, this condition covers all weighted Banzhaf values and Beta Shapley values with $\alpha, \beta \geq 1$.*

All in all, we demonstrate that by sticking to the principle of maximum sample reuse and avoiding any amplifying factors, we are able to establish a generic estimator that achieves the currently best convergence rate for any previously-studied semi-value.

## 4.3 A Connection between Probabilistic Values and Datamodels

A datamodel, proposed by Ilyas et al. (2022), is to learn an easy-to-interpret surrogate to represent a model output distribution related to a specific test example. In this circumstance, the set of players $[n]$ is identified with all the available training data. Precisely, the feature coordinates $\boldsymbol{\theta}^* \in \mathbb{R}^n$ imputed to every data point in $[n]$ is defined to be the uniquely optimal solution (together with a bias $b^* \in \mathbb{R}$) to the optimization problem

$$\operatorname*{argmin}_{\boldsymbol{\theta}\in\mathbb{R}^n, b\in\mathbb{R}} \sum_{S\subseteq[n]} \eta_{s+1} \left( U(S) - b - \sum_{i\in S}\theta_i \right)^2 \qquad (6)$$

where $\boldsymbol{\eta} \in \mathbb{R}^{n+1}$ is non-negative and $\sum_{s=0}^{n} \eta_{s+1} > 0$. The weight vector $\boldsymbol{\eta}$ can be scaled such that the objective in the problem (6) can be treated as an expectation, and thus the objective can be approximated through sampling a sufficient number of subsets, upon which an estimate of $(\boldsymbol{\theta}^*, b^*)$ can be obtained. We show below that $\boldsymbol{\theta}^*$ to a family of such least square regressions can be cast as some probabilistic values if it is the pairwise differences $\theta_j^* - \theta_k^*$ (for every $j, k \in [n]$) that matter.

**Theorem 2.** *Let $(b^*, \boldsymbol{\theta}^*)$ be the uniquely optimal solution to the problem (6) where $\eta_s = p_{s-1} + p_s$ for $2 \leq s \leq n$. Then, there is*

$$\theta_j^* - \theta_k^* = \phi_j - \phi_k \ \text{for every} \ j, k \in [n].$$

In other words, $\boldsymbol{\theta}^* = \boldsymbol{\phi} + c\mathbf{1}$; $\mathbf{1} \in \mathbb{R}^n$ is the all-one vector. When using datamodels to detect similar training examples to a given target, what matters is the relative order of components in $\boldsymbol{\theta}^*$. Meanwhile, Ilyas et al. (2022) showed that the corresponding performance depends on the choice of the weight vector $\boldsymbol{\eta}$. Therefore, our OFA-A estimator serves as a sufficient proxy for a range of $\{\boldsymbol{\theta}^*\}$ and would facilitate the fine-tuning of $\boldsymbol{\eta}$ when using datamodels to detect similar training examples.

**When $\boldsymbol{\theta}^*$ Can Be Recovered From $\boldsymbol{\phi}$** Interestingly, for specific choices of $\mathbf{p} \in \mathbb{R}^n$ and $\boldsymbol{\eta} \in \mathbb{R}^{n+1}$, it holds that $\boldsymbol{\theta} = \boldsymbol{\phi}$. Theorem 2 can be seen as an extension to the previous result stated in the below.

**Proposition 5** (Marichal and Mathonet 2011, Proposition 4)**.** *Suppose $0 < a < 1$ is given, if $p_j = a^{j-1}(1-a)^{n-j}$ for $1 \leq j \leq n$ and $\eta_k = a^{k-2}(1-a)^{n-k}$ for $1 \leq k \leq n+1$, which leads to $\eta_s = p_{s-1} + p_s$ for $2 \leq s \leq n$, there is*

$$\boldsymbol{\theta}^* = \boldsymbol{\phi}.$$

It is worth pointing out that $\boldsymbol{\phi}$ in Proposition 5 is exactly the weighted Banzhaf value parameterized by $a$, i.e., WB-$a$. Even more, under the same setting, we can even solve a group of datamodels with $\ell_1$ or $\ell_2$ regularization simultaneously.

**Corollary 1.** *Under the setting of Proposition 5, let $\boldsymbol{\theta}^*$ be the uniquely optimal solution to*

$$\underset{\boldsymbol{\theta} \in \mathbb{R}^n, b \in \mathbb{R}}{\operatorname{argmin}} \left( \sum_{S \subseteq [n]} \eta_{s+1} \left( U(S) - b - \sum_{i \in S} \theta_i \right)^2 \right) + \frac{\lambda}{a(1-a)} \mathcal{R}(\boldsymbol{\theta}) \qquad (7)$$

*where $\lambda > 0$, the following are true about the relation between $\boldsymbol{\theta}^*$ and $\boldsymbol{\phi}$:*

1. *If $\mathcal{R}(\boldsymbol{\theta}) = \|\boldsymbol{\theta}\|_2^2$, then*

$$\boldsymbol{\theta}^* = \left( 1 + \frac{\lambda}{a(1-a)} \right)^{-1} \boldsymbol{\phi}.$$

2. *If $\mathcal{R}(\boldsymbol{\theta}) = \|\boldsymbol{\theta}\|_1$, then*

$$\boldsymbol{\theta}^* = \operatorname{sign}(\boldsymbol{\phi}) \cdot \max \left( 0, |\boldsymbol{\phi}| - \frac{\lambda}{2a(1-a)} \right).$$

*All operations are element-wise.*

This corollary is immediate by combining Proposition 5, and Theorem 2.2 by Saunshi et al. (2022). We comment that replacing $x_i$ by $2x_i - 1$, i.e., mapping 0 and 1 into $-1$ and 1, respectively, in $\phi_{\{i\}}(x)$ used by Saunshi et al. (2022) yields $v_{\{i\}}(x)$ used by Marichal and Mathonet (2011). A remarkable implication of the combination of Corollary 1 and our proposed OFA-A estimator is that we can solve a group of regularized datamodels covered by the problem (7) *simultaneously*! For example, the coefficient $\lambda$ can be finetuned by running Algorithm 1 just once.

## 5 Experiments

In this section, we are to verify i) the simultaneous efficiency of our OFA-A estimator and ii) the faster convergence rate of our OFA-S estimator compared with the considered baselines and our OFA-A estimator. Particularly, if $D(\mathbf{m}, \mathbf{q})$ is effective in determining the convergence rate of Algorithm 1, our OFA-S estimator is expected to be faster than our OFA-A estimator. All the experiments are conducted using CPUs.

We use two types of utility functions for this end: i) following the experiment settings of (Li and Yu 2024), $U(S)$ is set to be the cross-entropy of LeNets trained on $S$ on the classification datasets FMNIST, MNIST and iris; to obtain the exact values, the number of training data $n$ is set to be 24; ii) $U$ is defined to be the sum of unanimity (SOU) games, i.e., $U(S) = \sum_{j=1}^d \alpha_j \mathbb{1}_{S_j \subseteq S}$ where each $\emptyset \subsetneq S_j \subsetneq [n]$ is randomly sampled, for which each semi-value can be computed by $\phi_i = \sum_{j=1}^d \alpha_j \int_{[0,1]} w^{s_j - 1} \mathrm{d}\mu(w)$; specifically, we set $n \in \{64, 128, 256\}$ with $d = n^2$, which implies that the implemented SOU games require $n^2$ utility evaluations to compute semi-values exactly. The random seed inside each utility function is fixed as 2024, and thus each $U$ is deterministic.

For the simplicity of presenting our empirical results, we use the area under the convergence curve (AUCC) to assess the convergence quality of estimators, and thus the smaller the better. For $n = 24$, the value of each player is approximated using $20,000$ utility evaluations, and we compute the AUCCs as $\frac{1}{100} \sum_{j=1}^{100} \frac{\|\hat{\phi}^{(200j)} - \phi\|_2}{\|\phi\|_2}$ where $\hat{\phi}^{(200j)}$ refers to the estimate using $200j$ utility evaluations for each player. For $n \in \{64, 128, 256\}$, the value of each player is approximated using $2,000$ utility evaluations, and the corresponding AUCCs are calculated as $\frac{1}{100} \sum_{j=1}^{100} \frac{\|\hat{\phi}^{(20j)} - \phi\|_2}{\|\phi\|_2}$. All the AUCCs are reported with standard deviation using 30 different random seeds from $\{0, 1, 2, \dots, 29\}$.

**Verification of Our OFA-A Estimator** For our OFA-A estimator where we substitute $\mathbf{q}^{\text{OFA-A}}$, which is defined in Proposition 1, into Algorithm 1, we choose the baselines according to Figure 1. The selected baselines include WSL-Shapley (Kwon and Zou 2022a), SHAP-IQ (Fumagalli et al. 2024), weightedSHAP (Kwon and Zou 2022b) and permutation-Shapley (Castro et al. 2009). The corresponding results are reported in Figure 2. Overall, our OFA-A estimator performs the best on all the employed 18 probabilistic values, which verify the simultaneous efficiency of our OFA-A estimator.

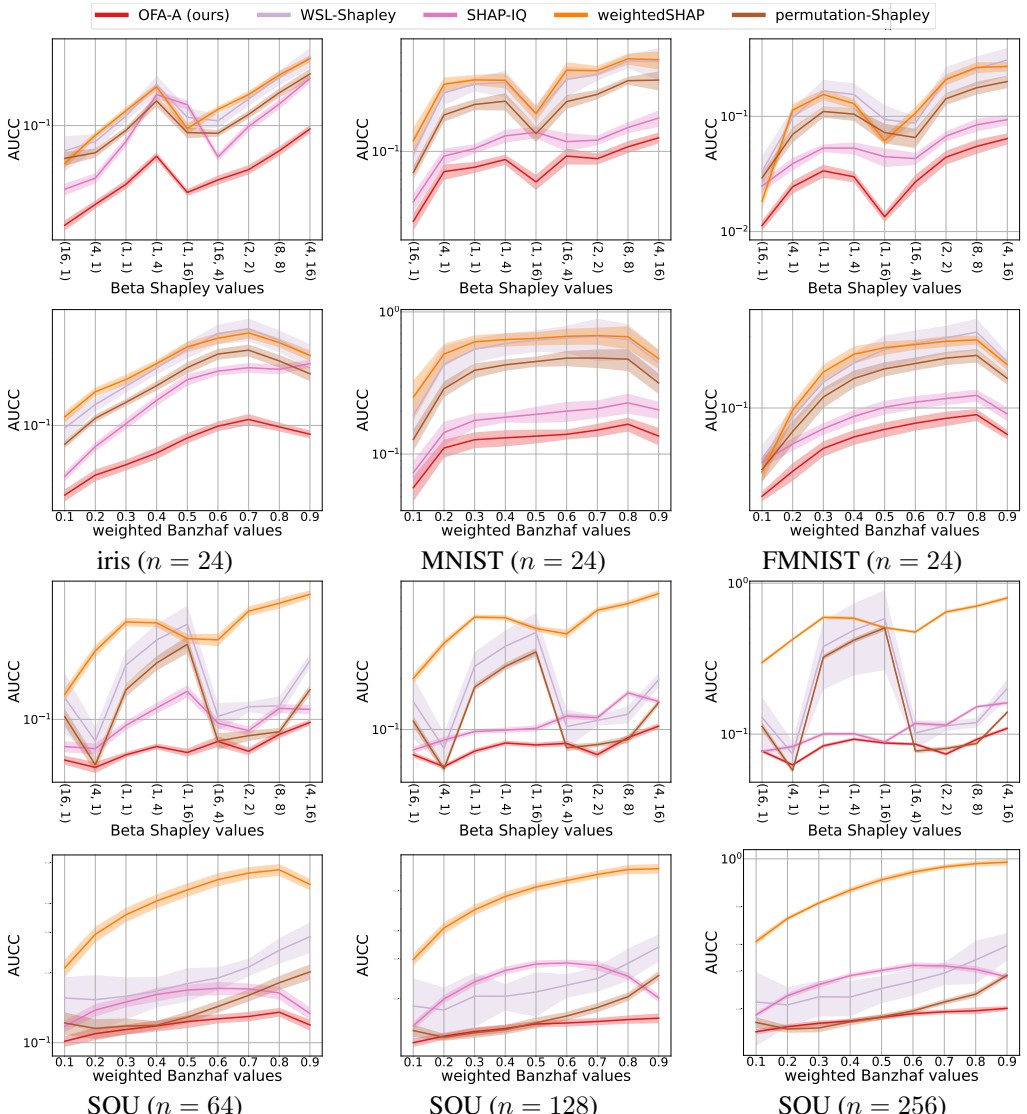

Figure 2: Comparison of one-for-all estimators using six utility functions. All the AUCCs are reported with standard deviation using 30 random seeds. Smaller AUCC indicates faster convergence rate.

**Verification of Our OFA-S Estimator** Next, we verify the faster convergence rate of our OFA-S estimator, using $\mathbf{q}^{\text{OFA-S}}$ as defined in Eq. (5). The baselines we employ include: (unbiased) kernelSHAP (Covert and Lee 2021; Lundberg and Lee 2017), GELS and GELS-Shapley (Li and Yu 2024), ARM (Kolpaczki et al. 2024; Li and Yu 2024), complement (Zhang et al. 2023), group testing (Jia et al. 2019; Wang and Jia 2023a), AME (Lin et al. 2022), MSR (Wang and Jia 2023b) and sampling lift (Moehle et al. 2022). Note that not all the baselines are designed for all the probabilistic values we employ. For example, the complement estimator only works for Beta$(1, 1)$, i.e., the Shapley value. According to (Li and Yu 2024, Remark 9), we implement the paired sampling technique for (unbiased) kernelSHAP and group testing. The corresponding results are presented in Figure 3.

First, our OFA-S estimator is indeed faster than our OFA-A estimator, which aligns exactly with our theory; in other words, it implies that our proposed $D(\mathbf{m}, \mathbf{p})$ indeed determines the convergence rate of our Algorithm 1. Second, our OFA-S estimator always performs the best except on the SOU games which require only $n^2$ utility evaluation to get the exact values; by contrast, the utility function defined using the classification datasets require $2^n$ utility evaluations instead. Third, our proposed estimator is consistently the fastest on the commonly-used Beta$(1, 1)$, i.e., the Shapley value; note that $\mathbf{q}^{\text{OFA-A}} = \mathbf{q}^{\text{OFA-S}}$ for the Shapley value; therefore, our proposed estimator achieves the currently best convergence rate both empirically and theoretically.

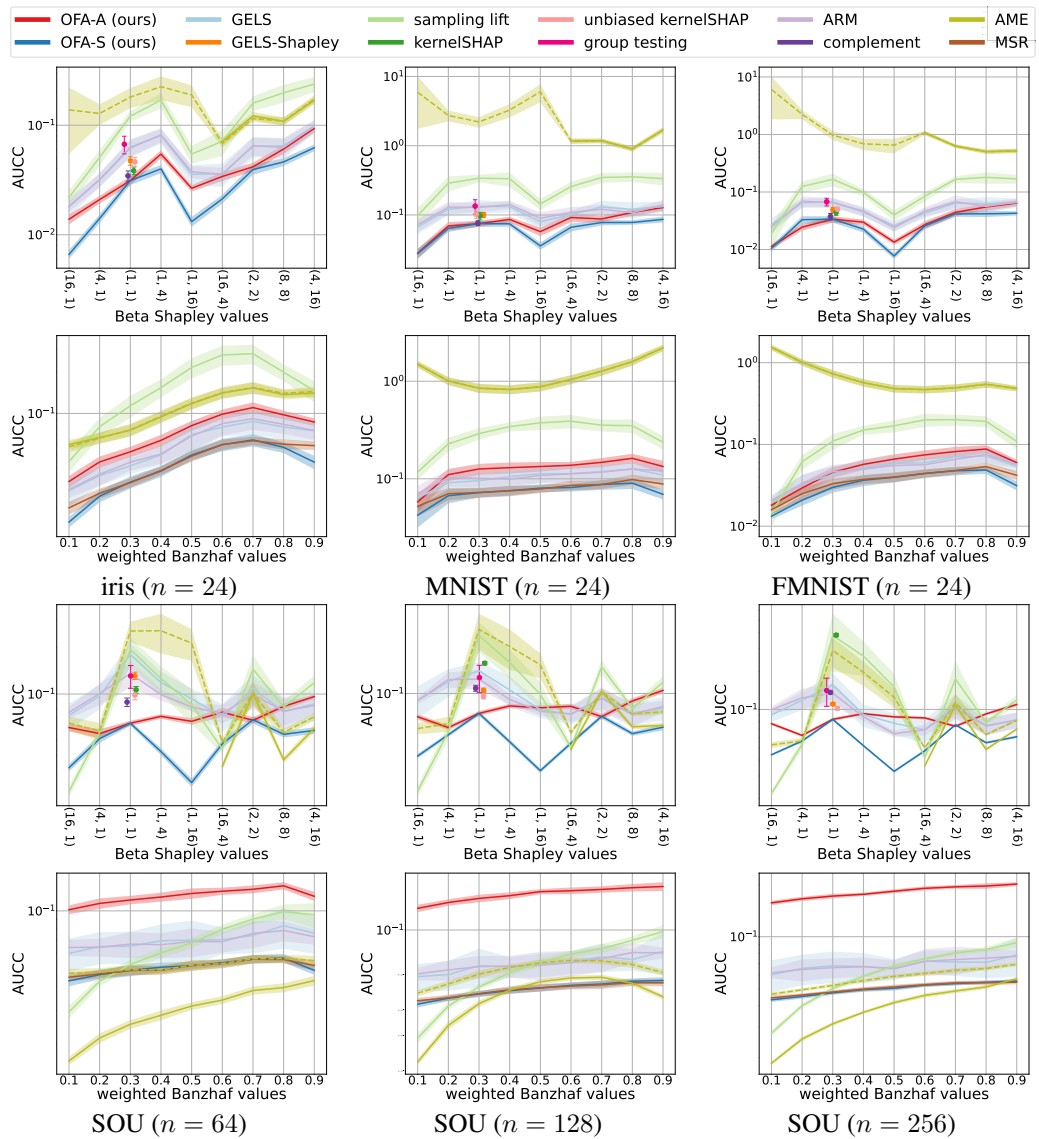

Figure 3: Comparison of twelve estimators using six utility functions. The dashed lines correspond to the improved AME estimator developed in Appendix D. All the AUCCs are reported with standard deviation using 30 random seeds. Smaller AUCC indicates faster convergence rate.

## 6 Conclusion

In this work, we propose a framework, termed OFA, that i) adheres to the principle of maximum sample reuse and ii) contains no amplifying factors for the goal of optimizing all probabilistic values simultaneously and efficiently. Particularly, our OFA framework is parameterized by a sampling vector $\mathbf{q} \in \mathbb{R}^{n-3}$. To gain insights, we theoretically develop a key formula $D(\mathbf{m}, \mathbf{q})$ concerning this framework that effectively determines the corresponding convergence rate. By optimizing $\mathbf{q}$ in $D(\mathbf{m}, \mathbf{q})$ for all probabilistic values on average, we obtain our one-for-all estimator that can theoretically approximate all probabilistic values simultaneously with the currently best convergence rate $O(n \log n)$ on average. Meanwhile, we propose a faster generic estimator by optimizing $\mathbf{q}$ for each specific probabilistic value, and we demonstrate that our generic estimate enjoys the best convergence rate for all previously-studied probabilistic values. All of our theoretical findings are verified in our experiments. Finally, we establish a connection between probabilistic values and the least square regressions used in datamodels, showing that our OFA-A estimator is capable of solving a family of (regularized) datamodels simultaneously.

# Acknowledgements

We thank the reviewers and the area chair for thoughtful comments that have improved our final presentation. YY gratefully acknowledges NSERC and CIFAR for funding support.

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

# A  Proof of Theorem 1

**Theorem 1.** *Assume i) $\|U\|_\infty \leq u$ and ii) $0 < \epsilon \leq \sqrt{2D(\mathbf{m}, \mathbf{q})\gamma(\mathbf{q})^2 u^2}$. For $\hat{\phi}$ in Algorithm 1, it requires $\frac{4nu^2 D(\mathbf{m},\mathbf{q})}{\epsilon^2} \log \frac{8n^2}{\delta}$ evaluations of $U$ to achieve $P(\|\hat{\phi} - \phi\|_2 \geq \epsilon) \leq \delta$ where*

$$D(\mathbf{m}, \mathbf{q}) = \sum_{s=2}^{n-2} \frac{n}{q_{s-1}} \left( \frac{m_s^2}{s} + \frac{m_{s+1}^2}{n-s} \right) \text{ and } \gamma(\mathbf{q}) = \min_{2 \leq s \leq n-2} \min \left( \frac{q_{s-1} \cdot s}{n}, \frac{q_{s-1} \cdot (n-s)}{n} \right).$$

*Proof.* Following Algorithm 1, let $\{S_t\}_{t=1}^T$ be $T$ independent random subsets. Define

$$T_{i,s}^+ = \sum_{t=1}^T [\![ i \in S_t, |S_t| = s ]\!] \text{ and } T_{i,s}^- = \sum_{t=1}^T [\![ i \notin S_t, |S_t| = s ]\!]$$

where $s = 2, 3, \ldots, n-2$. Then, we have

$$\hat{\phi}_{i,s}^+ = \frac{1}{T_{i,s}^+} \sum_{i=1}^T [\![ i \in S_t, |S_t| = s ]\!] \cdot U(S_t) \text{ and } \hat{\phi}_{i,s}^- = \frac{1}{T_{i,s}^-} \sum_{i=1}^T [\![ i \notin S_t, |S_t| = s ]\!] \cdot U(S_t).$$

Define $r_{i,s}^+ = \frac{T_{i,s}^+}{T}$ and $r_{i,s}^- = \frac{T_{i,s}^-}{T}$. In particular, both $[\![ i \in S_t, |S_t| = s ]\!]$ and $[\![ i \notin S_t, |S_t| = s ]\!]$ are Bernoulli random variables with

$$\mathbb{E}[r_{i,s}^+] = q_{s-1} \binom{n-1}{s-1} \binom{n}{s}^{-1} = \frac{q_{s-1} \cdot s}{n} \text{ and } \mathbb{E}[r_{i,s}^-] = q_{s-1} \binom{n-1}{s} \binom{n}{s}^{-1} = \frac{q_{s-1} \cdot (n-s)}{n}.$$

Additionally, $\mathbf{R}$ and $\boldsymbol{\tau}$ are defined to be vectors in $\mathbb{R}^{2n-6}$ such that $R_{2k-1} = r_{i,k+1}^+$, $R_{2k} = r_{i,k+1}^-$, $\tau_{2k-1} = \frac{q_k \cdot (k+1)}{n}$ and $\tau_{2k} = \frac{q_k \cdot (n-k-1)}{n}$ for $k \in [n-3]$. Note that $\mathbf{R}$ is a random vector. By Hoeffding's inequality,

$$P(|R_j - \tau_j| \geq \omega) \leq 2 \exp \left( -2T\omega^2 \right)$$

where $\omega > 0$, and thus

$$P(\|\mathbf{R} - \boldsymbol{\tau}\|_\infty \geq \omega) \leq P\left( \bigcup_{j \in [2n-6]} |R_j - \tau_j| \geq \omega \right) \leq (4n - 12) \exp \left( -2T\omega^2 \right).$$

Denote the event $\{ \sum_{s=2}^{n-2} [m_s(\hat{\phi}_{i,s}^+ - \phi_{i,s}^+) + m_{s+1}(\phi_{i,s}^- - \hat{\phi}_{i,s}^-)] \geq \epsilon \}$ by $E_i$ where $\epsilon > 0$. Let $\mathcal{C}$ be the set that contains all possible configurations $\mathbf{C} \in \{0,1\}^{(2n-6) \times T}$ such that $\frac{\mathbf{C1}_T}{T} = \mathbf{R}$ and $\mathbf{1}_T^\top \mathbf{C} = \mathbf{1}_{2n-6}^\top$, i.e., $C_{j,k} = 0$ indicates that the $k$-th subset is sampled from $\{R \subseteq [n] \mid r = (j+3)/2 \text{ and } i \in R\}$ if $j$ is odd and $\{R \subseteq [n] \mid r = (j+2)/2 \text{ and } i \notin R\}$ otherwise. Then,

$$P(E_i) = \sum_{\mathbf{C} \in \mathcal{C}} P(E_i \cap \mathbf{C}) = \sum_{\mathbf{C} \in \mathcal{C}} P(E_i \mid \mathbf{C}) \cdot P(\mathbf{C}).$$

Observe that $\mathcal{C}$ can be divided into two separate groups $\mathcal{C}_{<\omega}$ and $\mathcal{C}_{\geq\omega}$ such that

$$\sum_{\mathbf{C}_{<\omega} \in \mathcal{C}_{<\omega}} P(\mathbf{C}_{<\omega}) = P(\|\mathbf{R} - \boldsymbol{\tau}\|_\infty < \omega) \text{ and } \sum_{\mathbf{C}_{\geq\omega} \in \mathcal{C}_{\geq\omega}} P(\mathbf{C}_{\geq\omega}) = P(\|\mathbf{R} - \boldsymbol{\tau}\|_\infty \geq \omega).$$

Therefore,

$$\begin{aligned}
P(E_i) &= \sum_{\mathbf{C}_{<\omega} \in \mathcal{C}_{<\omega}} P(E_i \mid \mathbf{C}_{<\omega}) \cdot P(\mathbf{C}_{<\omega}) + \sum_{\mathbf{C}_{\geq\omega} \in \mathcal{C}_{\geq\omega}} P(E_i \mid \mathbf{C}_{\geq\omega}) \cdot P(\mathbf{C}_{\geq\omega}) \\
&\leq \sum_{\mathbf{C}_{<\omega} \in \mathcal{C}_{<\omega}} P(E_i \mid \mathbf{C}_{<\omega}) \cdot P(\mathbf{C}_{<\omega}) + (4n - 12) \exp \left( -2T\omega^2 \right).
\end{aligned} \tag{8}$$

For simplicity, we write $P_{\mathbf{C}_{<\omega}}(E_i)$ instead of $P(E_i \mid \mathbf{C}_{<\omega})$. Additionally, we assume $\omega < \frac{\gamma(\mathbf{q})}{2}$ so that neither $T_{i,s}^+$ nor $T_{i,s}^-$ is zero when conditioned on any $\mathbf{C}_{<\omega}$. By the Chernoff bound, for any

$\lambda > 0$, there is

$$P_{\mathbf{C}_{<\omega}}(E_i) \leq \mathbb{E}_{\mathbf{C}_{<\omega}}\left[\exp\left(\lambda \sum_{s=2}^{n-2}\left(m_s(\hat{\phi}_{i,s}^+ - \phi_{i,s}^+) + m_{s+1}(\phi_{i,s}^- - \hat{\phi}_{i,s}^-)\right)\right)\right] \cdot e^{-\lambda\epsilon}$$

$$= e^{-\lambda\epsilon} \prod_{s=2}^{n-2} \mathbb{E}_{\mathbf{C}_{<\omega}}\left[\exp\left(\lambda m_s(\hat{\phi}_{i,s}^+ - \phi_{i,s}^+)\right)\right] \prod_{s=2}^{n-2} \mathbb{E}_{\mathbf{C}_{<\omega}}\left[\exp\left(\lambda m_{s+1}(\phi_{i,s}^- - \hat{\phi}_{i,s}^-)\right)\right]$$

where the equality is due to the independence that stems from the independence of random subsets and that the configuration is fixed. Moreover,

$$\mathbb{E}_{\mathbf{C}_{<\omega}}\left[\exp\left(\lambda m_s(\hat{\phi}_{i,s}^+ - \phi_{i,s}^+)\right)\right] = \mathbb{E}_{\mathbf{C}_{<\omega}}\left[\exp\left(\lambda m_s \frac{1}{T_{i,s}^+}\sum_{j=1}^{T_{i,s}^+}(U(S_{i,s,j}^+) - \phi_{i,s}^+)\right)\right]$$

$$= \prod_{j=1}^{T_{i,s}^+} \mathbb{E}_{\mathbf{C}_{<\omega}}\left[\exp\left(\frac{\lambda m_s}{T_{i,s}^+}(U(S_{i,s,j}^+) - \phi_{i,s}^+)\right)\right]$$

where $\{S_{i,s,j}^+\}_{1 \leq j \leq T_{i,s}^+}$ is obtained by ordering $\{S_t \mid |S_t| = s \text{ and } i \in S_t\}$. In a similar fashion, we have

$$\mathbb{E}_{\mathbf{C}_{<\omega}}\left[\exp\left(\lambda m_{s+1}(\phi_{i,s}^- - \hat{\phi}_{i,s}^-)\right)\right] = \prod_{j=1}^{T_{i,s}^-} \mathbb{E}_{\mathbf{C}_{<\omega}}\left[\exp\left(\frac{\lambda m_{s+1}}{T_{i,s}^-}(\phi_{i,s}^- - U(S_{i,s,j}^-))\right)\right]$$

By Hoeffding's lemma,

$$\mathbb{E}_{\mathbf{C}_{<\omega}}\left[\exp\left(\frac{\lambda m_s}{T_{i,s}^+}(U(S_{i,s,j}^+) - \phi_{i,s}^+)\right)\right] \leq \exp\left(\frac{\lambda^2 m_s^2 u^2}{2T_{i,s}^+ \cdot T_{i,s}^+}\right),$$

$$\mathbb{E}_{\mathbf{C}_{<\omega}}\left[\exp\left(\frac{\lambda m_{s+1}}{T_{i,s}^-}(\phi_{i,s}^- - U(S_{i,s,j}^-))\right)\right] \leq \exp\left(\frac{\lambda^2 m_{s+1}^2 u^2}{2T_{i,s}^- \cdot T_{i,s}^-}\right),$$

which leads to

$$\prod_{j=1}^{T_{i,s}^+} \mathbb{E}_{\mathbf{C}_{<\omega}}\left[\exp\left(\frac{\lambda m_s}{T_{i,s}^+}(U(S_{i,s,j}^+) - \phi_{i,s}^+)\right)\right] \leq \exp\left(\frac{\lambda^2 m_s^2 u^2}{2T_{i,s}^+}\right),$$

$$\prod_{j=1}^{T_{i,s}^-} \mathbb{E}_{\mathbf{C}_{<\omega}}\left[\exp\left(\frac{\lambda m_{s+1}}{T_{i,s}^-}(\phi_{i,s}^- - U(S_{i,s,j}^-))\right)\right] \leq \exp\left(\frac{\lambda^2 m_{s+1}^2 u^2}{2T_{i,s}^-}\right).$$

Therefore,

$$P_{\mathbf{C}_{<\omega}}(E_i) \leq \exp\left(\frac{\lambda^2 u^2}{2T}\hat{D} - \lambda\epsilon\right)$$

where $\hat{D} = \sum_{s=2}^{n-2}\left(\frac{T}{T_{i,s}^+}m_s^2 + \frac{T}{T_{i,s}^-}m_{s+1}^2\right)$. Next, we aim to show that $|\hat{D} - D(\mathbf{m}, \mathbf{q})| \leq D(\mathbf{m}, \mathbf{q})$. Observe that

$$|\hat{D} - D(\mathbf{m}, \mathbf{q})| \leq \sum_{s=2}^{n-2}\left(\left|\frac{1}{r_{2s-3}} - \frac{1}{\tau_{2s-3}}\right|m_s^2 - \left|\frac{1}{r_{2s-2}} - \frac{1}{\tau_{2s-2}}\right|m_{s+1}^2\right),$$

and since $|r_j - \tau_j| < \omega$,

$$\left|\frac{1}{r_j} - \frac{1}{\tau_j}\right| \leq \frac{\omega}{(\tau_j - \omega)\tau_j} = \frac{1}{\tau_j - \omega} - \frac{1}{\tau_j}.$$

Since $\gamma(\mathbf{q}) \leq \tau_j$ and $\omega \leq \frac{\gamma(\mathbf{q})}{2}$,

$$\frac{1}{\tau_j - \omega} = \frac{\tau_j}{\tau_j - \omega} \cdot \frac{1}{\tau_j} = \frac{1}{1 - \frac{\omega}{\tau_j}} \cdot \frac{1}{\tau_j} \leq \frac{2}{\tau_j}.$$

As a result, we have $|\hat{D} - D(\mathbf{m}, \mathbf{q})| \le D(\mathbf{m}, \mathbf{q})$, and thus

$$P_{\mathbf{C}_{<\omega}}(E_i) \le \exp\left(\frac{\lambda^2 u^2}{T}D(\mathbf{m}, \mathbf{q}) - \lambda\epsilon\right). \tag{9}$$

Combining Eqs. (8) and (9) yields

$$P(E_i) \le \exp\left(\frac{\lambda^2 u^2}{T}D(\mathbf{m}, \mathbf{q}) - \lambda\epsilon\right) + (4n - 12)\exp(-2T\omega^2).$$

Choosing $\lambda > 0$ that minimizes the upper bound yields

$$P(E_i) \le \exp\left(-\frac{T\epsilon^2}{4u^2 D(\mathbf{m}, \mathbf{q})}\right) + (4n - 12)\exp(-2T\omega^2).$$

Solving the equation $-\frac{T\epsilon^2}{4u^2 D(\mathbf{m},\mathbf{q})} = -2T\omega^2$ yields $\omega = \sqrt{\frac{\epsilon^2}{8D(\mathbf{m},\mathbf{q})u^2}}$, which gives

$$-2T\omega^2 = -\frac{T\epsilon^2}{4D(\mathbf{m}, \mathbf{q})u^2}.$$

Particularly, to meet the assumption $\omega \le \frac{\gamma(\mathbf{q})}{2}$, we have to have $\epsilon \le \sqrt{2D(\mathbf{m}, \mathbf{q})\gamma(\mathbf{q})^2 u^2}$. To conclude, provided that $\epsilon \le \sqrt{2D(\mathbf{m}, \mathbf{q})\gamma(\mathbf{q})^2 u^2}$, we have

$$P(\sum_{s=2}^{n-2}\left(m_s(\hat{\phi}_{i,s}^+ - \phi_{i,s}^+) + m_{s+1}(\phi_{i,s}^- - \hat{\phi}_{i,s}^-)\right) \ge \epsilon) \le 4n\exp(-\frac{T\epsilon^2}{4D(\mathbf{m}, \mathbf{q})u^2}).$$

Similarly, there is

$$P(\sum_{s=2}^{n-2}\left(m_s(\phi_{i,s}^+ - \hat{\phi}_{i,s}^+) + m_{s+1}(\hat{\phi}_{i,s}^- - \phi_{i,s}^-)\right) \ge \epsilon) \le 4n\exp(-\frac{T\epsilon^2}{4D(\mathbf{m}, \mathbf{q})u^2}),$$

and thus

$$P(|\hat{\phi}_i - \phi_i| \ge \epsilon) \le 8n\exp(-\frac{T\epsilon^2}{4D(\mathbf{m}, \mathbf{q})u^2}).$$

Eventually, we have

$$P(\|\hat{\boldsymbol{\phi}} - \boldsymbol{\phi}\|_2 \ge \epsilon) \le P(\bigcup_{i\in[n]} |\hat{\phi}_i - \phi_i| \ge \frac{\epsilon}{\sqrt{n}}) \le 8n^2\exp(-\frac{T\epsilon^2}{4nD(\mathbf{m}, \mathbf{q})u^2}).$$

Solving $\delta \ge 8n^2\exp(-\frac{T\epsilon^2}{4nD(\mathbf{m},\mathbf{q})u^2})$ yields $T \ge \frac{4nD(\mathbf{m},\mathbf{q})u^2}{\epsilon^2}\log\frac{8n^2}{\delta}$. Note the assumption $\epsilon \le \sqrt{2D(\mathbf{m}, \mathbf{q})\gamma(\mathbf{q})^2 u^2}$ can be removed if the configuration is fixed with $T_{i,s}^+ \approx \frac{s \cdot q_{s-1}}{n}T$ and $T_{i,s}^- \approx \frac{(n-s)q_{s-1}}{n}T$. $\square$

# B   Proofs of Propositions

**Proposition 1.** $q_{s-1}^{OFA\text{-}A} \propto \frac{1}{\sqrt{(s)(n-s)}}$ *and* $\overline{D}(\mathbf{q}^{OFA\text{-}A}) \in O(1)$. *In other words, our OFA-A estimator achieves the convergence rate of $O(n\log n)$ simultaneously for all probabilistic values on average.*

*Proof.* Let $\Lambda = \{\mathbf{x} \in \mathbb{R}^{n-1} \mid 0 \le \sum_{j=1}^{n-1} x_j \le L_n\}$ where $L_n = n^{\frac{1}{2(n-1)}}$, and a smooth homeomorphism $f : \Lambda \to \Delta$ is defined by letting

$$f(\mathbf{x}) = \frac{1}{L_n}(x_1, x_2, \cdots, x_{n-1}, L_n - \sum_{j=1}^{n-1} x_j)^\top.$$

In other words, both $f$ and $f^{-1}$ are $C^\infty$. Since the volume of $\Delta$ is $\frac{n^{\frac{1}{2}}}{(n-1)!}$, there is

$$\frac{(n-1)!}{n^{\frac{1}{2}}} \int_{\mathbf{x}\in\Lambda} D(f(\mathbf{x}),\mathbf{q})\sqrt{\det\left(Df(\mathbf{x})^\top Df(\mathbf{x})\right)}\mathrm{d}\mathbf{x} = \int_{\mathbf{m}\in\Delta} D(\mathbf{m},\mathbf{q})\mathrm{d}\nu(\mathbf{m}).$$

Note that $\sqrt{\det\left(Df(\mathbf{x})^\top Df(\mathbf{x})\right)} = 1$ for every $\mathbf{x}\in\Lambda$. With $\overline{\Lambda} = \{\mathbf{y}\in\mathbb{R}^{n-1} \mid 0 \le \sum_{j=1}^{n-1} y_j \le 1\}$, we have

$$\frac{(n-1)!}{n^{\frac{1}{2}}} \int_{\mathbf{x}\in\Lambda} D(f(\mathbf{x}),\mathbf{q})\mathrm{d}\mathbf{x} = (n-1)! \int_{\mathbf{y}\in\overline{\Lambda}} D(f(L_n\mathbf{y}),\mathbf{q})\mathrm{d}\mathbf{y}.$$

For simplicity, assume that $n=4$, notice that

$$\int_{y\in\overline{\Lambda}} y_{n-1}^2 \mathrm{d}\mathbf{y} = \int_0^1 \mathrm{d}y_1 \int_0^{1-y_1} \mathrm{d}y_2 \int_0^{1-y_1-y_2} y_3^2 \mathrm{d}y_3 = \frac{1}{3\cdot4\cdot5} = \frac{1}{\prod_{k=1}^{n-1}(2+k)}.$$

Therefore,

$$\int_{\mathbf{y}\in\overline{\Lambda}} D(f(L_n\mathbf{y}),\mathbf{q})\mathrm{d}\mathbf{y} = \sum_{s=2}^{n-2} \frac{n}{q_{s-1}} \int_{y\in\overline{\Lambda}} \left(\frac{y_s^2}{s} + \frac{y_{s+1}^2}{n-s}\right) \mathrm{d}\mathbf{y}$$

$$= \frac{1}{\prod_{k=1}^{n-1}(2+k)} \sum_{s=2}^{n-2} \frac{n}{q_{s-1}} \left(\frac{1}{s} + \frac{1}{n-s}\right),$$

which leads to

$$\overline{D}(\mathbf{q}) = \frac{(n-1)!}{\prod_{k=1}^{n-1}(2+k)} \sum_{s=2}^{n-2} \frac{n}{q_{s-1}} \left(\frac{1}{s} + \frac{1}{n-s}\right).$$

Since $\overline{D}(\mathbf{q})$ is convex in $\mathbf{q}$, $\mathbf{q}^{\text{OFA-A}}$ can be directly obtained using the KKT conditions, which is

$$q_{s-1}^{\text{OFA-A}} = \frac{\sqrt{\frac{n}{s} + \frac{n}{n-s}}}{\sum_{s=2}^{n-2} \sqrt{\frac{n}{s} + \frac{n}{n-s}}}.$$

Therefore, we have

$$\overline{D}(\mathbf{q}^{\text{OFA-A}}) = \frac{(n-1)!}{\prod_{k=1}^{n-1}(2+k)} \left(\sum_{s=2}^{n-2} \sqrt{\frac{n}{s} + \frac{n}{n-s}}\right)^2.$$

Since $\lim_{n\to\infty} \frac{(n-1)!(n-1)^2}{\prod_{k=1}^{n-1}(2+k)} = 2\Gamma(3)$, when $n$ is sufficiently large, there is

$$\overline{D}(\mathbf{q}^{\text{OFA-A}}) \approx \frac{1}{n^2} \left(\sum_{s=2}^{n-2} \sqrt{\frac{n}{s} + \frac{n}{n-s}}\right)^2 = \left(\frac{1}{n}\sum_{s=2}^{n-2} \sqrt{\frac{1}{\frac{s}{n}\left(1-\frac{s}{n}\right)}}\right)^2 < \left(\int_0^1 \frac{1}{x(1-x)}\mathrm{d}x\right)^2 = \pi^2.$$

$\square$

**Proposition 2.** *Our OFA-A estimator achieves the convergence rate of $O(n\log n)$ simultaneously for all semi-values whose probability density functions exist and are bounded. Particularly, Beta Shapley values with $\alpha, \beta \ge 1$ all satisfy this condition.*

*Proof.* Let $\phi$ be a semi-value such that $p_s = \int_0^1 w^{s-1}(1-w)^{n-s}\mathrm{d}\mu(w) = \int_0^1 w^{s-1}(1-w)^{n-s}p_\mu(w)\mathrm{d}w$ such that $p_\mu(w) \le B$ for every $w \in [0,1]$. Particularly, we have

$$m_s = \binom{n-1}{s-1} p_s \le B \cdot \binom{n-1}{s-1} \int_0^1 w^{s-1}(1-w)^{n-s}\mathrm{d}w = B \cdot \binom{n-1}{s-1}\frac{(s-1)!(n-s)!}{n!} = \frac{B}{n}.$$

Therefore,

$$D(\mathbf{m},\mathbf{q}^{\text{OFA-A}}) \le \frac{B^2}{n} \sum_{s=2}^{n-2} \frac{1}{q_{s-1}^{\text{OFA-A}}} \left(\frac{1}{s} + \frac{1}{n-s}\right) = B^2 \left(\sum_{s=2}^{n-2} \frac{1}{\sqrt{s(n-s)}}\right)^2 < B^2\pi^2.$$

$\square$

**Proposition 3.** *If $p_s = a^{s-1}(1-a)^{n-s}$ with $0 < a < 1$, which corresponds to the weighted Banzhaf value parameterized by $w$, then $D(\mathbf{m}, \mathbf{q}^{OFA\text{-}A}) \in O(n^{\frac{1}{2}})$. In other words, the OFA estimator achieves the convergence rate of $O(n^{\frac{3}{2}} \log n)$ simultaneously for all WB-$a$ with $0 < a < 1$.*

*Proof.* With $q_{s-1}^{\text{OFA-A}} \propto \frac{1}{\sqrt{s(n-s)}}$, we have

$$D(\mathbf{m}, \mathbf{q}^{\text{OFA-A}}) = C \cdot n \cdot \sum_{s=2}^{n-2} \left( \sqrt{\frac{n-s}{s}} m_s^2 + \sqrt{\frac{s}{n-s}} m_{s+1}^2 \right) \text{ where } C = \sum_{s=2}^{n-2} \frac{1}{\sqrt{s(n-s)}} < \pi$$

Then,

$$D(\mathbf{m}, \mathbf{q}^{\text{OFA-A}})$$

$$= C \sum_{s=2}^{n-2} n \left( \sqrt{\frac{n-s}{s}} \binom{n-1}{s-1}^2 \left(a^{s-1}(1-a)^{n-s}\right)^2 + \sqrt{\frac{s}{n-s}} \binom{n-1}{s}^2 \left(a^s(1-a)^{n-s-1}\right)^2 \right).$$

Specifically,

$$\sqrt{\frac{n-s}{s}} \binom{n-1}{s-1}^2 = \sqrt{\frac{s}{n-s}} \frac{(n-1)!^2}{(s-1)!s!(n-s-1)!(n-s)!}$$

$$\text{and } \sqrt{\frac{s}{n-s}} \binom{n-1}{s}^2 = \sqrt{\frac{n-s}{s}} \frac{(n-1)!^2}{(s-1)!s!(n-s-1)!(n-s)!},$$

and thus

$$n \cdot \left( \sqrt{\frac{n-s}{s}} \binom{n-1}{s-1}^2 \left(a^{s-1}(1-a)^{n-s}\right)^2 + \sqrt{\frac{s}{n-s}} \binom{n-1}{s}^2 \left(a^s(1-a)^{n-s-1}\right)^2 \right)$$

$$= n \cdot \left(a^{s-1}(1-a)^{n-s-1}\right)^2 \frac{(n-1)!^2}{(s-1)!s!(n-s-1)!(n-s)!} \left( \sqrt{\frac{s}{n-s}}(1-a)^2 + \sqrt{\frac{n-s}{s}} a^2 \right)$$

Since

$$\sqrt{\frac{s}{n-s}}(1-a)^2 + \sqrt{\frac{n-s}{s}} a^2 \le \sqrt{\frac{s}{n-s}} + \sqrt{\frac{n-s}{s}} = \frac{n}{\sqrt{s(n-s)}},$$

there is

$$n \cdot \left( \sqrt{\frac{n-s}{s}} \binom{n-1}{s-1}^2 \left(a^{s-1}(1-a)^{n-s}\right)^2 + \sqrt{\frac{s}{n-s}} \binom{n-1}{s}^2 \left(a^s(1-a)^{n-s-1}\right)^2 \right)$$

$$\le \sqrt{s(n-s)} \frac{\left(\binom{n}{s} a^s (1-a)^{n-s}\right)^2}{a^2(1-a)^2} \le n \cdot \frac{\left(\binom{n}{s} a^s (1-a)^{n-s}\right)^2}{a^2(1-a)^2}.$$

Using the identity $\sum_{j=0}^m \binom{m}{j}^2 (x+y)^{2j}(x-y)^{2(m-j)} = \sum_{j=0}^m \binom{2j}{j}\binom{2(m-j)}{m-j} x^{2j} y^{2(m-j)}$, there is

$$\sum_{s=2}^{n-2} \left( \binom{n}{s} a^s (1-a)^{n-s} \right)^2 = \sum_{s=0}^{n} \binom{2s}{s}\binom{2(n-s)}{n-s} \frac{1}{2^{2s}} \left( \frac{2a-1}{2} \right)^{2(n-s)}$$

$$= \binom{2n}{n} \left( \frac{2a-1}{2} \right)^{2n} + \sum_{s=1}^{n-1} \binom{2s}{s}\binom{2(n-s)}{n-s} \frac{1}{2^{2s}} \left( \frac{2a-1}{2} \right)^{2(n-s)} + \binom{2n}{n} \frac{1}{2^{2n}}.$$

For every $k \ge 1$, $\binom{2k}{k} \approx \frac{2^{2k}}{\sqrt{k}}$ using the Stirling's approximation, and thus

$$\binom{2n}{n} \left( \frac{2a-1}{2} \right)^{2n} \approx \frac{z^n}{\sqrt{n}}, \quad \binom{2n}{n} \frac{1}{2^{2n}} \approx \frac{1}{\sqrt{n}}$$

$$\sum_{s=1}^{n-1} \binom{2s}{s}\binom{2(n-s)}{n-s} \frac{1}{2^{2s}} \left( \frac{2a-1}{2} \right)^{2(n-s)} \approx \sum_{s=1}^{n-1} \frac{1}{\sqrt{s(n-s)}} z^{n-s} \le \frac{\sum_{j=1}^{n-1} z^j}{\sqrt{n-1}},$$

where $z = (2a-1)^2 < 1$. Therefore, we obtain $\sum_{s=2}^{n-2} \left(\binom{n}{s}a^s(1-a)^{n-s}\right)^2 \leq O(n^{-\frac{1}{2}})$, which eventually leads to

$$D(\mathbf{m}, \mathbf{q}^{\text{OFA-A}}) \leq \frac{n}{a^2(1-a)^2} \sum_{s=2}^{n-2} \left(\binom{n}{s}a^s(1-a)^{n-s}\right)^2 \leq O(n^{\frac{1}{2}}).$$

$\square$

**Proposition 4.** *For semi-values, $D(\mathbf{m}, \mathbf{q}^{OFA\text{-}S}) \in O(1)$ if i) $\mu$ has a bounded probability density function or ii) $\int_{(0,1)} \frac{1}{w(1-w)}\mathrm{d}\mu(w) < \infty$. Particularly, this condition covers all weighted Banzhaf values and Beta Shapley values with $\alpha, \beta \geq 1$.*

*Proof.* If $\mu(\{0\}) \neq 0$ ($\mu(\{1\}) \neq 0$, respectively), its induced marginal contributions all reside in $\phi_{i,1}^+$ and $\phi_{i,0}^-$ ($\phi_{i,n}^+$ and $\phi_{i,n-1}^-$, respectively), which is computed exactly using Algorithm 1. Therefore, W.L.O.G., we assume that $\mu((0,1)) = 1$.

Suffice it to show that if $\int_0^1 \frac{1}{w(1-w)}\mathrm{d}\mu(w) < \infty$, there is

$$\sum_{s=2}^{n-2} \sqrt{\frac{n}{s}m_s^2 + \frac{n}{n-s}m_{s+1}^2} \in O(1).$$

Specifically,

$$\mathbf{D}(\mathbf{m}, \mathbf{q}^{\text{OFA-S}}) = \sum_{s=2}^{n-2} \sqrt{\frac{n}{s}m_s^2 + \frac{n}{n-s}m_{s+1}^2} \leq \sum_{s=2}^{n-2} \left(\sqrt{\frac{n}{s}}m_s + \sqrt{\frac{n}{n-s}}m_{s+1}\right)$$

$$= \int_0^1 \sum_{s=2}^{n-2} \left(\sqrt{\frac{s}{n}}\binom{n}{s}w^{s-1}(1-w)^{n-s} + \sqrt{\frac{n-s}{n}}\binom{n}{s}w^s(1-w)^{n-s-1}\right)\mathrm{d}\mu(w).$$

Since $\sqrt{\frac{s}{n}}(1-w) + \sqrt{\frac{n-s}{n}}w \leq 2$, we have

$$\int_0^1 \sum_{s=2}^{n-2} \left(\sqrt{\frac{s}{n}}\binom{n}{s}w^{s-1}(1-w)^{n-s} + \sqrt{\frac{n-s}{n}}\binom{n}{s}w^s(1-w)^{n-s-1}\right)\mathrm{d}\mu(w)$$

$$\leq \int_0^1 \sum_{s=2}^{n-2} \frac{2\binom{n}{s}w^s(1-w)^{n-s}}{w(1-w)}\mathrm{d}\mu(w) \leq 2\int_0^1 \frac{1}{w(1-w)}\mathrm{d}\mu(w) \in O(1).$$

$\square$

## C  Proof of Theorem 2

To prove this theorem, we first state useful definitions and lemmas.

**Definition 2** (Semi Inner Product). *Let $\mathcal{V}$ is a real linear space. A semi inner product $\langle \cdot, \cdot \rangle$ on $\mathcal{V}$ satisfies, for every $x, y, z \in \mathcal{V}$ and every $\alpha \in \mathbb{R}$, i) $\langle x, y \rangle = \langle y, x \rangle$, ii) $\langle \alpha x, y \rangle = \alpha\langle x, y \rangle$, iii) $\langle x + y, z \rangle = \langle x, z \rangle + \langle y, z \rangle$, and iv) $\langle x, x \rangle \geq 0$. In addition, we write $\|x\| = \sqrt{\langle x, x \rangle}$ for every $x \in \mathcal{V}$.*

**Lemma 1.** *Let a semi inner product on a linear space $\mathcal{V}$ be given, and $\mathcal{A} \subseteq \mathcal{V}$ is some affine space. For the following optimization problem*

$$\underset{x \in \mathcal{A}}{\arg\min} \|x - p\|^2$$

*where $p \in \mathcal{V}$, $x^*$ is optimal if and only if*

$$\langle x^* - p, y - x^* \rangle = 0, \ \forall y \in \mathcal{A}. \tag{10}$$

*Proof.* Suppose $x^*$ verifies Eq. (10), for every $y \in \mathcal{A}$,

$$\|y - p\|^2 = \|x^* - p\|^2 + \|y - x^*\|^2 + 2\langle x^* - p, y - x^* \rangle \geq \|x^* - p\|^2.$$

Next, suppose $x^*$ is optimal, and for the sake of contradiction, assume that there is some $y \in \mathcal{A}$ such that $\langle x^* - p, y - x^* \rangle \neq 0$. Write $z = y - x^*$, for $t \in \mathbb{R}$

$$\|x^* + tz - p\|^2 = \|x^* - p\|^2 + t^2 \|z\|^2 + 2t\langle x^* - p, z \rangle.$$

Since $\langle x^* - p, z \rangle \neq 0$, there exists some $t_o \in \mathbb{R}$ such that $t^2 \|z\|^2 + 2t\langle x^* - p, z \rangle < 0$, and thus $\|x^* + t_o z - p\|^2 < \|x^* - p\|^2$, a contradiction. □

**Definition 3** (Projection Induced by a Semi Inner Product). *Given a semi inner product on a linear space $\mathcal{V}$, the set of all optimal solutions to the problem*

$$\underset{x \in \mathcal{A}}{\operatorname{argmin}} \|x - p\|^2,$$

*where $\mathcal{A} \subseteq \mathcal{V}$ is an affine space and $p \in \mathcal{V}$, is denoted by $\operatorname{Proj}_{\mathcal{A}}(\{p\})$. To account for the possibility that there are multiple optimal solutions, we extend the definition by letting $\operatorname{Proj}_{\mathcal{A}}(S) = \bigcup_{p \in S} \operatorname{Proj}_{\mathcal{A}}(\{p\})$.*

**Lemma 2.** *Let $\mathcal{V}$ be a linear space with a semi inner product. Suppose there are two affine spaces $\mathcal{B} \subseteq \mathcal{A}$, for every $p \in \mathcal{V}$, there is*

$$\operatorname{Proj}_{\mathcal{B}}(\operatorname{Proj}_{\mathcal{A}}(\{p\})) \subseteq \operatorname{Proj}_{\mathcal{B}}(\{p\}).$$

*Proof.* We rephrase Lemma 1 to ease the proof. For each affine space $\mathcal{A} \subseteq \mathcal{V}$, define $\mathcal{L}_{\mathcal{A}} = \mathcal{A} - q$ for some $q \in \mathcal{A}$. Note that the resulting $\mathcal{L}_{\mathcal{A}}$ is independent of the choice of $q \in \mathcal{A}$ and it is a subspace in $\mathcal{V}$. Therefore, Eq. (10) is equivalent to

$$\langle x^* - p, z \rangle = 0, \ \forall z \in \mathcal{L}_{\mathcal{A}}.$$

Suppose $x \in \operatorname{Proj}_{\mathcal{B}}(\operatorname{Proj}_{\mathcal{A}}(\{p\}))$, by Lemma 1, there exists $y \in \operatorname{Proj}_{\mathcal{A}}(\{p\})$ such that

$$\langle y - p, a \rangle = 0, \ \forall a \in \mathcal{L}_{\mathcal{A}} \ \text{and} \ \langle x - y, b - x \rangle = 0, \ \forall b \in \mathcal{B}.$$

Therefore,

$$\langle x - p, b - x \rangle = \langle x - y, b - x \rangle + \langle y - p, b - x \rangle = 0 + 0.$$

$\langle y - p, b - x \rangle = 0$ is due to that $b - x \in \mathcal{L}_{\mathcal{B}} \subseteq \mathcal{L}_{\mathcal{A}}$. □

**Lemma 3** (Ruiz et al. 1998, Theorem 12). *Let $\mathbf{v}^*$ be the uniquely optimal solution to*

$$\underset{\mathbf{v} \in \mathbb{R}^n}{\operatorname{argmin}} \sum_{S \subseteq [n]} \eta_{s+1} \left( U(S) - U(\emptyset) - \sum_{i \in S} v_i \right)^2 \quad \text{s.t.} \quad \sum_{i \in [n]} v_i = U([n]) - U(\emptyset) \qquad (11)$$

*where $\eta_s = p_{s-1} + p_s$ for $2 \leq s \leq n$. Then, there is*

$$v_i^* - v_j^* = \phi_i - \phi_j \ \text{for every } i, j \in [n].$$

Recall that the problem (6) is

$$\underset{\boldsymbol{\theta} \in \mathbb{R}^n, b \in \mathbb{R}}{\operatorname{argmin}} \sum_{S \subseteq [n]} \eta_{s+1} \left( U(S) - b - \sum_{i \in S} \theta_i \right)^2,$$

and our goal is to prove that

$$\theta_i^* - \theta_j^* = v_i^* - v_j^* \ \text{for every} \ i, j \in [n],$$

which together with Lemma 3 is sufficient to complete our proof.

**Theorem 2.** *Let $(b^*, \boldsymbol{\theta}^*)$ be the uniquely optimal solution to the problem (6) where $\eta_s = p_{s-1} + p_s$ for $2 \leq s \leq n$. Then, there is*

$$\theta_j^* - \theta_k^* = \phi_j - \phi_k \ \text{for every} \ j, k \in [n].$$

*Proof.* The first part of our proof was inspired by (Hammer and Holzman 1992, Lemma 2.9). Let $\mathcal{G} = \{U : 2^{[n]} \to \mathbb{R}\}$, $\mathcal{AG} = \{U \in \mathcal{G} \mid U(S) = a_0 + \sum_{i \in S} a_i \text{ for every } S \subseteq [n]\}$ and $\mathcal{A}_U = \{g \in \mathcal{AG} \mid U([n]) = g([n]) \text{ and } U(\emptyset) = g(\emptyset)\}$. Note that $\mathcal{G}$ is a linear space and the other two are affine spaces with $\mathcal{A}_U \subseteq \mathcal{AG}$. For clarity, each game in $\mathcal{AG}$ is written as $[a_0, \mathbf{a}]$ where $\mathbf{a} \in \mathbb{R}^n$.

A semi inner product on $\mathcal{G}$ can be defined by letting $\langle g_1, g_2 \rangle = \sum_{S \subseteq [n]} \eta_{s+1} \cdot g_1(S) g_2(S)$ for every $g_1, g_2 \in \mathcal{G}$. Then, $[b^*, \boldsymbol{\theta}^*]$ is the projection of $U$ onto $\mathcal{AG}$, whereas $[U(\emptyset), \mathbf{v}^*]$ is the projection of $U$ onto $\mathcal{A}_U$ where $\mathbf{v}^*$ is the uniquely optimal solution to the problem (11).

By Lemma 2, there is $\text{Proj}_{\mathcal{A}_U}(\text{Proj}_{\mathcal{AG}}(\{U\})) \subseteq \text{Proj}_{\mathcal{A}_U}(\{U\})$. Moreover, the uniqueness in problem (11) implies that $\text{Proj}_{\mathcal{A}_U}(\{U\}) = \{[U(\emptyset), \mathbf{v}^*]\}$, and thus

$$\text{Proj}_{\mathcal{A}_U}(\text{Proj}_{\mathcal{AG}}(\{U\})) = \text{Proj}_{\mathcal{A}_U}(\{U\}) = \{[U(\emptyset), \mathbf{v}^*]\}.$$

Since $[b^*, \boldsymbol{\theta}^*] \in \text{Proj}_{\mathcal{AG}}(\{U\})$, the equality $\text{Proj}_{\mathcal{A}_U}(\{[b^*, \boldsymbol{\theta}^*]\}) = \{[U(\emptyset), \mathbf{v}^*]\}$ means that $[U_\emptyset, \mathbf{v}^*]$ is the uniquely optimal solution to the problem

$$\underset{[U(\emptyset), \mathbf{v}] \in \mathcal{A}_U}{\text{argmin}} \sum_{S \subseteq [n]} \eta_{s+1} \left([U(\emptyset), \mathbf{v}](S) - [b^*, \boldsymbol{\theta}^*](S)\right)^2. \tag{12}$$

Pick $i, j \in [n]$ such that $i \neq j$, and define an additive game $\mathbf{e}^i \in \mathcal{AG}$ by letting $\mathbf{e}^i(S) = 1$ if $i \in S$ and 0 otherwise, $\mathbf{e}^j$ is defined similarly. Consider the problem

$$\underset{t \in \mathbb{R}}{\text{argmin}} \sum_{S \subseteq [n]} \eta_{s+1} \left([U(\emptyset), \mathbf{v}^*](S) - [b^*, \boldsymbol{\theta}^*](S) + t(\mathbf{e}^i(S) - \mathbf{e}^j(S))\right)^2. \tag{13}$$

Note that $[U(\emptyset), \mathbf{v}^*] + t(\mathbf{e}^i - \mathbf{e}^j) \in \mathcal{A}_U$ for every $t \in \mathbb{R}$, and the uniqueness to the problem (12) suggests that $t^* = 0$ is the uniquely optimal solution to the problem (13). Removing all constant terms in the problem (13) yields an equivalent problem

$$\underset{t \in \mathbb{R}}{\text{argmin}} \sum_{S \subseteq [n]:\, i \in S, j \notin S} \eta_{s+1} \left([U(\emptyset), \mathbf{v}^*](S) - [b^*, \boldsymbol{\theta}^*](S) + t\right)^2$$
$$+ \sum_{S \subseteq [n]:\, i \notin S, j \in S} \eta_{s+1} \left([U(\emptyset), \mathbf{v}^*](S) - [b^*, \boldsymbol{\theta}^*](S) - t\right)^2.$$

Write $g = [U(\emptyset), \mathbf{v}^*] - [b^*, \boldsymbol{\theta}^*]$, since this problem is convex, letting the derivative equal 0 leads to

$$t^* = \frac{\sum_{S \subseteq [n]:\, i \notin S, j \in S} \eta_{s+1} \cdot g(S) - \sum_{S \subseteq [n]:\, i \in S, j \notin S} \eta_{s+1} \cdot g(S)}{2 \sum_{S:\, i \in S, j \notin S} \eta_{s+1}} = 0.$$

Write $g = [g_0, \mathbf{g}]$ where $g_0 = U(\emptyset) - b^*$ and $\mathbf{g} = \mathbf{v}^* - \boldsymbol{\theta}^*$, there is

$$\sum_{S \subseteq [n]:\, i \in S, j \notin S} \eta_{s+1} \cdot g(S) = \alpha(g_0 + g_i) + \beta \sum_{1 \leq k \leq n:\, k \neq i, j} g_k$$

$$\text{where } \alpha = \sum_{s=1}^{n-1} \binom{n-2}{s-1} \eta_{s+1} \text{ and } \beta = \sum_{s=2}^{n-1} \binom{n-3}{s-2} \eta_{s+1}.$$

Similarly, we have $\sum_{S \subseteq [n]:\, i \notin S, j \in S} \eta_{s+1} \cdot g(S) = \alpha(g_0 + g_j) + \beta \sum_{1 \leq k \leq n:\, k \neq i, j} g_k$, and therefore

$$\alpha(g_j - g_i) = 0.$$

Since $\alpha > 0$, we eventually get $g_i = g_j$. In other words, $v_i^* - \theta_i^* = v_j^* - \theta_j^*$. Because $i$ and $j$ are chosen arbitrarily, our proof is completed.

To be self-contained, we also prove that the problem (6) has only one optimal solution provided that $\eta_s = p_{s-1} + p_s$ for $2 \leq s \leq n$. W.L.O.G., assume $\sum_{S \subseteq [n]} \eta_{s+1} = 1$. By letting the derivative of the

problem (6) equal 0, we have $\mathbf{A}\mathbf{x} = \mathbf{b}$

$$\mathbf{A} = \begin{pmatrix} 1 & \kappa & \kappa & \cdots & \kappa \\ \kappa & \kappa & \tau & \cdots & \tau \\ \kappa & \tau & \kappa & \ddots & \vdots \\ \vdots & \vdots & \ddots & \ddots & \tau \\ \kappa & \tau & \cdots & \tau & \kappa \end{pmatrix},$$

$$\kappa = \sum_{s=1}^{n} \binom{n-1}{s-1}\eta_{s+1}, \quad \tau = \sum_{s=2}^{n} \binom{n-2}{s-2}\eta_{s+1}, \quad b_1 = \sum_{S \subseteq [n]} \eta_{s+1}U(S),$$

$$b_{j+1} = \sum_{S \subseteq [n]:\, j \in S} \eta_{s+1}U(S) \text{ for every } j \in [n], \quad x_1 = b \text{ and } x_{j+1} = \theta_j \text{ for every } j \in [n].$$

Left multiplying $\mathbf{A}$ with some row operation matrix $\mathbf{R}$ gives

$$\mathbf{R}\mathbf{A} = \begin{pmatrix} 1 & \kappa & \kappa & \cdots & \kappa \\ 0 & \kappa - \kappa^2 & \tau - \kappa^2 & \cdots & \tau - \kappa^2 \\ 0 & \tau - \kappa^2 & \kappa - \kappa^2 & \ddots & \vdots \\ \vdots & \vdots & \ddots & \ddots & \ddots \\ 0 & \tau - \kappa^2 & \cdots & \tau - \kappa^2 & \kappa - \kappa^2 \end{pmatrix}.$$

It is sufficient to prove that the bottom-right $n \times n$ submatrix of $\mathbf{R}\mathbf{A}$ is invertible. Suffice it to show $\kappa - \tau \neq 0$ and $\kappa + (n-1)\tau - n\kappa^2 \neq 0$. Using $\binom{n}{s} = \binom{n-1}{s} + \binom{n-1}{s-1}$, we have

$$\kappa - \tau = \sum_{s=1}^{n-1} \binom{n-2}{s-1}\eta_{s+1} > 0.$$

Using $n\binom{n-1}{s-1} = s\binom{n}{s}$, we have

$$\kappa + (n-1)\tau = \sum_{s=1}^{n} s\binom{n-1}{s-1}\eta_{s+1} = \frac{1}{n}\sum_{s=1}^{n} s^2\binom{n}{s}\eta_{s+1},$$

$$n \cdot \kappa^2 = n \cdot \left(\sum_{s=1}^{n} \binom{n-1}{s-1}\eta_{s+1}\right)^2 = \frac{1}{n}\left(\sum_{s=1}^{n} s\binom{n}{s}\eta_{s+1}\right)^2.$$

Let $\gamma = 1 - \eta_1$ and $\zeta_s = \eta_{s+1}/\gamma$ for every $s \in [n]$, there is

$$n \cdot \kappa^2 = \frac{\gamma^2}{n}\left(\sum_{s=1}^{n} s\binom{n}{s}\zeta_s\right)^2 = \frac{\gamma^2}{n}\mathbb{E}[s]^2 \leq \frac{\gamma^2}{n}\mathbb{E}[s^2] = \gamma(\kappa + (n-1)\tau) \leq \kappa + (n-1)\tau.$$

If $\eta_1 > 0$, the last inequality is strict as $\gamma < 1$. Otherwise, the first inequality is strict as $\mathrm{Var}[s] = \mathbb{E}[s^2] - \mathbb{E}[s]^2 > 0$. $\qquad \square$

## D  Overview of Estimators

Recall that each probabilistic value is defined to be, for every $i \in [n]$,

$$\phi_i = \phi_i(U) = \sum_{S \subseteq [n]\setminus i} p_{s+1}(U(S \cup i) - U(S)) \tag{14}$$

where $\mathbf{p} \in \mathbb{R}^n$ is a non-negative vector with $\sum_{s=1}^{n} \binom{n-1}{s-1}p_s = 1$. If $p_s = \int_0^1 w^{s-1}(1-w)^{n-s}\mathrm{d}\mu(w)$ for some probability measure $\mu$ on the closed interval $[0,1]$, the induced $\phi$ is referred to as a semi-value.

**The Sampling Lift Estimator (Moehle et al. 2022)** The sampling lift estimator is based on

$$\phi_i = \mathbb{E}_{S \subseteq [n] \setminus i}[U(S \cup i) - U(S)] \quad \text{where} \quad P(S) = p_{s+1}.$$

The sampling procedure is: i) sample a subset size $s \in [n]$ with $P(s) = \binom{n-1}{s-1}p_s$, and then ii) sample a subset $S$ uniformly from $\{R \subseteq [n] \setminus i \mid r = s - 1\}$. For semi-values such that $p_s = \int_0^1 w^{s-1}(1-w)^{n-s}\mathrm{d}\mu(w)$ where $\mu$ is a probability measure on the closed interval $[0,1]$, there is an alternative: i) sample a $w \in [0,1]$ according to $\mu$, and then sample a subset $S \subseteq [n] \setminus i$ by incorporating each player in $[n] \setminus i$ with probability $w$. With a sequence of sampled subsets $\{S_j\}_{j=1}^T$, the $i$-th estimate is $\hat{\phi}_i = \frac{1}{T}\sum_{j=1}^T (U(S_j \cup i) - U(S_j))$.

**The Weighted Sampling Lift Estimator (Kwon and Zou 2022a)** The formula it is built upon is

$$\phi_i = \mathbb{E}_{S \subseteq [n] \setminus i}^{\text{Shap}}\left[\frac{p_{s+1}}{q_{s+1}}(U(S \cup i) - U(S))\right] \quad \text{where} \quad P(S) = q_{s+1} = \frac{s!(n-1-s)!}{n!}.$$

Note that setting $\mathbf{p} = \mathbf{q}$ in Eq. (14) leads to the Shapley value. The sampling procedure is: i) sample a $w$ uniformly from $[0,1]$, and then ii) sample a subset $S \subseteq [n] \setminus i$ by incorporating each player in $[n] \setminus i$ with probability $w$. Then, the $i$-th estimate is $\hat{\phi}_i = \frac{1}{T}\sum_{j=1}^T \frac{p_{s_j+1}}{q_{s_j+1}}(U(S_j \cup i) - U(S_j))$.

**The KernelSHAP Estimator (Lundberg and Lee 2017)** This estimator is specific to the Shapley value. It employs the fact that the Shapley value $\phi_i^{\text{Shap}}$ is the uniquely optimal solution to

$$\operatorname*{argmin}_{\boldsymbol{\phi} \in \mathbb{R}^n} \sum_{\emptyset \subsetneq S \subsetneq [n]} \binom{n-2}{s-1}^{-1} \left(U(S) - U(\emptyset) - \sum_{i \in S}\phi_i\right)^2 \quad \text{s.t.} \quad \sum_{i \in [n]}\phi_i = U([n]) - U(\emptyset). \quad (15)$$

Note that the weights can be scaled so that the objective is an expectation. A sequence of subsets $\{S_j\}_{j=1}^T$ where $\emptyset \subsetneq S_j \subsetneq [n]$ is sampled according to $P(S) \propto \binom{n-2}{s-1}^{-1}$. Then, we have an approximate problem as

$$\operatorname*{argmin}_{\boldsymbol{\phi} \in \mathbb{R}^n} \frac{1}{T}\sum_{j=1}^T \left(U(S_j) - U(\emptyset) - \sum_{i \in S_j}\phi_i\right)^2 \quad \text{s.t.} \quad \sum_{i \in [n]}\phi_i = U([n]) - U(\emptyset),$$

the uniquely optimal solution of which is treated as the estimates, i.e.,

$$\hat{\boldsymbol{\phi}}^{\text{Shap}} = \hat{\mathbf{A}}^{-1}\left(\hat{\mathbf{b}} - \mathbf{1}_n \frac{\mathbf{1}_n^\top \hat{\mathbf{A}}^{-1}\hat{\mathbf{b}} - U([n]) + U(\emptyset)}{\mathbf{1}_n^\top \hat{\mathbf{A}}^{-1}\mathbf{1}_n}\right)$$

$$\text{where} \quad \hat{\mathbf{A}} = \frac{1}{T}\sum_{j=1}^T \mathbf{1}_{S_j}\mathbf{1}_{S_j}^\top \quad \text{and} \quad \hat{\mathbf{b}} = \frac{1}{T}\sum_{j=1}^T (U(S_j) - U(\emptyset)) \cdot \mathbf{1}_{S_j}.$$

Specifically, $\mathbf{1}_{S_j} \in \{0,1\}^n$ such that its $i$-th entry is 1 if and only if $i \in S_j$.

**The Unbiased KernelSHAP Estimator (Covert and Lee 2021)** The uniquely optimal solution $\phi^{\text{Shap}}$ to the problem (15) is

$$\phi^{\text{Shap}} = \mathbf{A}^{-1}\left(\mathbf{b} - \mathbf{1}_n \frac{\mathbf{1}_n^\top \mathbf{A}^{-1}\mathbf{b} - U([n]) + U(\emptyset)}{\mathbf{1}_n^\top \mathbf{A}^{-1}\mathbf{1}_n}\right)$$

$$\text{where} \quad \mathbf{A} = \mathbb{E}[\mathbf{1}_S \mathbf{1}_S^\top] \quad \text{and} \quad \mathbf{b} = \mathbb{E}[(U(S) - U(\emptyset)) \cdot \mathbf{1}_n].$$

This estimator employs the fact that $\mathbf{A}_{ij} = \frac{1}{2}$ if $i = j$ and $\frac{1}{n(n-1)}\frac{\sum_{s=2}^{n-1}\frac{s-1}{n-s}}{\sum_{s=1}^{n-1}\frac{1}{s(n-s)}}$ otherwise. In other words, the estimates of this estimator is

$$\hat{\boldsymbol{\phi}}^{\text{Shap}} = \mathbf{A}^{-1}\left(\hat{\mathbf{b}} - \mathbf{1}_n \frac{\mathbf{1}_n^\top \mathbf{A}^{-1}\hat{\mathbf{b}} - U([n]) + U(\emptyset)}{\mathbf{1}_n^\top \mathbf{A}^{-1}\mathbf{1}_n}\right) \quad \text{where} \quad \hat{\mathbf{b}} = \frac{1}{T}\sum_{j=1}^T (U(S_j) - U(\emptyset)) \cdot \mathbf{1}_{S_j}.$$

$$(16)$$

Particularly $\{S_j\}_{j=1}^T$ where $\emptyset \subsetneq S_j \subsetneq [n]$ are sampled using $P(S) \propto \binom{n-2}{s-1}^{-1}$.

Recently, Fumagalli et al. (2024) proved that Eq. (16) can be simplified as

$$\hat{\phi}_i^{\text{Shap}} = \frac{U([n]) - U(\emptyset)}{n} + \frac{2\sum_{s=1}^{n-1}\frac{1}{s}}{T}\sum_{j=1}^T U(S_j)\left(\mathbb{1}_{i\in S_j} - \frac{s_j}{n}.\right)$$

**The ARM Estimator (Kolpaczki et al. 2024)**   This estimator is designed according to

$$\phi_i = \mathbb{E}_{S\sim P^+|i\in S}[U(S)] - \mathbb{E}_{S\sim P^-|i\notin S}[U(S)]$$

where $P^+(S) \propto p_s$ for every $\emptyset \subsetneq S \subseteq [n]$ and $P^-(S) \propto p_{s+1}$ for every $S \subsetneq [n]$ (Li and Yu 2024, Proposition 8). A sequence of subsets $\{S_j\}_{j=1}^T$ are sampled using $P^+$ and $P^-$ alternatively, i.e., $\{S_{2k-1}\}_{k=1}^{\frac{T}{2}}$ are sampled independently according to $P^+$, whereas $\{S_{2k}\}_{k=1}^{\frac{T}{2}}$ are sampled independently using $P^-$. Then, the $i$-th estimate is

$$\hat{\phi}_i = \frac{1}{T_i^+}\sum_{k=1}^{\frac{T}{2}} U(S_{2k-1})\mathbb{1}_{i\in S_{2k-1}} - \frac{1}{T_i^-}\sum_{k=1}^{\frac{T}{2}} U(S_{2k})\mathbb{1}_{i\notin S_{2k}}$$

where $T_i^+ = \sum_{k=1}^{\frac{T}{2}}\mathbb{1}_{i\in S_{2k-1}}$ and $T_i^- = \sum_{k=1}^{\frac{T}{2}}\mathbb{1}_{i\notin S_{2k}}$.

**The AME Estimator (Lin et al. 2022)**   This estimator is restricted to a sub-family of semi-values that satisfy $\int_0^1 \frac{1}{w(1-w)}\mathrm{d}\mu(w) < \infty$. For such a semi-value $\phi$, it can be cast as a uniquely optimal solution to

$$\underset{\mathbf{v}\in\mathbb{R}^n}{\arg\min}\,\mathbb{E}[(Y - \boldsymbol{X}^\top\mathbf{v})^2]$$

where $\boldsymbol{X} \in \mathbb{R}^n$ and $Y$ are random variables. The sampling procedure is: i) sample a $w \in (0,1)$ using $\mu$, ii) sample a subset $S$ by incorporating each player with probability $w$, and then iii) $Y = U(S)$ and $\boldsymbol{X} = \boldsymbol{X}(S)$ such that $X_i = \frac{1}{w\cdot C}$ if $i \in S$ and $-\frac{1}{(1-w)C}$ otherwise where $C = \int_0^1 \frac{1}{w(1-w)}\mathrm{d}\mu(w)$. With a sequence of subsets $\{(w_j, S_j)\}_{j=1}^T$, the uniquely optimal solution to the approximate problem

$$\underset{\mathbf{v}\in\mathbb{R}^n}{\arg\min}\,\frac{1}{T}\sum_{j=1}^T \left(U(S_j) - \boldsymbol{X}(S_j)^\top\mathbf{v}\right)^2$$

is taken as the induced estimates, which is $\hat{\boldsymbol{\phi}} = (\mathbf{A}^\top\mathbf{A})^{-1}\mathbf{A}^\top\mathbf{b}$ where the $j$-th row of $\mathbf{A}$ is $\boldsymbol{X}(S_j)^\top$ and $b_j = U(S_j)$.

**One Way to Improve the AME Estimator**   The limitation of the AME estimator is that it only applies to semi-values that satisfy $\int_0^1 \frac{1}{w(1-w)}\mathrm{d}\mu(w) < \infty$. Meanwhile, another potential drawback is its need to compute the inverse of $\mathbf{A}^\top\mathbf{A}$, though it can be circumvented by solving the approximate problem using gradients. In this work, we make a small improvement to the AME estimator by extending its applicability to all semi-values, removing $(\mathbf{A}^\top\mathbf{A})^{-1}$ in the approximate formula and providing a more direct analysis of its convergence rate in terms of $(\epsilon, \delta)$-approximation.

Our improvement begins with the observation that $\mathbb{E}[\boldsymbol{X}\boldsymbol{X}^\top] = \mathbf{I}$, suggesting that $\left(\frac{1}{T}\mathbf{A}^\top\mathbf{A}\right)^{-1} \to \mathbf{I}$ by the law of large numbers and thus $(\mathbf{A}^\top\mathbf{A})^{-1}$ is redundant. Its removal leads to a simplified formula:

$$\hat{\phi}_i = \frac{1}{T}\sum_{j=1}^T \left(\frac{[\![i\in S_j]\!]}{w_j}U(S_j) - \frac{[\![i\notin S_j]\!]}{1-w_j}U(S_j)\right).$$

We comment that the following proposition is complementary to (Lin et al. 2022, Proposition 3.3) that claims a similar result.

**Proposition 6.** *Assume that i) $\|U\|_\infty \leq u$ and ii) $\mu([A,B]) = 1$ for some $0 < A < B < 1$, the improved AME estimator requires $\frac{2nu^2C^2}{\epsilon^2}\log\frac{2n}{\delta}$ utility evaluations of $U$ to achieve $P(\|\hat{\boldsymbol{\phi}} - \boldsymbol{\phi}\|_2 \geq \epsilon) \leq \delta$ where $C = \frac{1}{\min(A, 1-B)}$.*

*Proof.* Notice that $\hat{\phi}_i = \frac{1}{T}\sum_{j=1}^{T} Z_j$ where $\{Z_j\}_{j=1}^{T}$ are i.i.d. random variables with $\mathbb{E}[Z_j] = \phi_i$. By the Hoeffding's inequality, there is

$$P(|\hat{\phi}_i - \phi_i| \geq \epsilon) \leq 2\exp\left(-\frac{T\epsilon^2}{2u^2C^2}\right)$$

where $C = \frac{1}{\min(A, 1-B)}$. Then,

$$P(\|\hat{\boldsymbol{\phi}} - \boldsymbol{\phi}\|_2 \geq \epsilon) \leq P(\bigcup_{1 \leq i \leq n} |\hat{\phi}_i - \phi_i| \geq \frac{\epsilon}{\sqrt{n}}) \leq 2n\exp\left(-\frac{T\epsilon^2}{2nu^2C^2}\right).$$

Solving $2n\exp\left(-\frac{T\epsilon^2}{2nu^2C^2}\right) \leq \delta$ leads to $T \geq \frac{2nu^2C^2}{\epsilon^2}\log\frac{2n}{\delta}$. $\qquad\square$

**Remark 1.** *Notice that the improved AME estimator requires that $\mu(\{0,1\}) = 0$. Nevertheless, semi-values are additively decomposable on $\mu$ and the part related to $\mu(\{0,1\})$ can be computed exactly in linear time. Therefore, it is fair to conclude that the improved AME estimator applies to all semi-values.*

**The MSR Estimator (Wang and Jia 2023b)** The methodology of this estimator is limited to weighted Banzhaf values parameterized with $0 < a < 1$ (Wang and Jia 2023b, Appendix C.2). Precisely, $p_s = a^{s-1}(1-a)^{n-s}$. Each subset is sampled by incorporating each player with probability $a$, and then the $i$-th estimate is

$$\hat{\phi}_i = \frac{1}{T_i^+}\sum_{j=1}^{T} U(S_j)\mathbb{1}_{i \in S_j} - \frac{1}{T_i^-}\sum_{j=1}^{T} U(S_j)\mathbb{1}_{i \notin S_j}$$

where $T_i^+ = \sum_{j=1}^{T}\mathbb{1}_{i \in S_j}$ and $T_i^- = \sum_{j=1}^{T}\mathbb{1}_{i \notin S_j}$.

**The GELS Estimator (Li and Yu 2024)** This estimator is established using the fact that $\phi_i = v_i^* - v_{n+1}^*$ where $\mathbf{v}^* \in \mathbb{R}^{n+1}$ is the uniquely optimal solution to

$$\underset{\mathbf{v} \in \mathbb{R}^{n+1}}{\arg\min} \sum_{\emptyset \subsetneq S \subsetneq [n+1]} p_s \left(U(S \cap [n]) - \sum_{i \in S} v_i\right)^2.$$

The subsets $\{S_j\}_{j=1}^{T}$ where $\emptyset \subsetneq S_j \subsetneq [n+1]$ are sampled using $P(S) \propto p_s$, and then the $i$-th estimate is

$$\hat{\phi}_i = \left(\sum_{s=1}^{n}\binom{n}{s-1}p_s\right)(\hat{v}_i - \hat{v}_{n+1})$$

where $\hat{v}_k = \frac{1}{T_k}\sum_{j=1}^{T} U(S_j \cap [n])\mathbb{1}_{k \in S_j}$ and $T_k = \sum_{j=1}^{T}\mathbb{1}_{k \in S_j}$.

**The Complement Estimator (Zhang et al. 2023)** The complement estimator is specific to the Shapley value using the fact that

$$\phi_i^{\text{Shap}} = \frac{1}{n}\sum_{S \subseteq [n]\backslash i}\binom{n-1}{s}^{-1}(U(S \cup i) - U([n]\backslash(S \cup i))).$$

The sequence of subsets $\{S_j\}_{j=1}^{T}$ is sampled using i) sample a subset size $s \in [n]$ uniformly, and then sample a subset $S$ uniformly from $\{R \subseteq [n] \mid r = s\}$. Then, the $i$-th estimate is

$$\hat{\phi}_i^{\text{Shap}} = \frac{1}{n}\sum_{s=1}^{n}\hat{\phi}_{i,s} \text{ where } \hat{\phi}_{i,s} = \frac{1}{T_{i,s}}\sum_{j=1}^{n}(v_j[\![i \in S_j, s_j = s]\!] - v_j[\![i \notin S_j, n - s_j = s]\!])$$

$$v_j = U(S_j) - U([n]\backslash S_j) \text{ and } T_{i,s} = \sum_{j=1}^{T}([\![i \in S_j, s_j = s]\!] + [\![i \notin S_j, n - s_j = s]\!]).$$

**The Group Testing Estimator (Jia et al. 2019)**   We introduce the improved version presented by Wang and Jia (2023a). Note that this estimator is specific to the Shapley value. A sequence of subsets $\{S_j\}_{j=1}^T$ are independently sampled according to: i) sample a subset size $s \in [n]$ using $P(s) \propto \frac{1}{s(n+1-s)}$, and then ii) sample a subset $S$ uniformly from $\{R \subseteq [n+1] \mid r = s\}$. Then, the $i$-th estimate is

$$\hat{\phi}_i^{\text{Shap}} = \frac{2 \sum_{s=1}^n \frac{1}{s}}{T} \sum_{j=1}^T U(S_j \cap [n]) \left( [\![ i \in S_j, n+1 \notin S_j ]\!] - [\![ i \notin S_j, n+1 \in S_j ]\!] \right).$$

**The Permutation Estimator (Castro et al. 2009)**   This estimator is specific to the Shapley value, using the formula

$$\phi_i^{\text{Shap}} = \frac{1}{n!} \sum_{\pi \in \Pi} (U(\mathcal{P}^i(\pi) \cup i) - U(\mathcal{P}^i(\pi)))$$

where $\Pi$ contains all permutations of $[n]$ and $\mathcal{P}^i(\pi)$ is the subset that contains all players preceding $i$ in $\pi$. Thus, it samples a sequence of permutations $\{\pi_j\}_{j=1}^T$ from $\Pi$ uniformly with replacement, and then the $i$-th estimate is $\hat{\phi}_i^{\text{Shap}} = \frac{1}{T} \sum_{j=1}^T \left( U(\mathcal{P}^i(\pi_j) \cup i) - U(\mathcal{P}^i(\pi_j)) \right)$.

**The WeightedSHAP Estimator (Kwon and Zou 2022b)**   As mentioned in the main paper, it is based on

$$\phi_i = \sum_{s=1}^n m_s \cdot \mathbb{E}_{\substack{R \subseteq [n] \setminus i \\ r = s-1}}[U(R \cup i) - U(R)]$$

where $m_s = \binom{n-1}{s-1} p_s$. For each player $i \in [n]$, it samples a sequence of permutations $\{\pi_j\}_{j=1}^T$ of $[n] \setminus i$. Then, the corresponding estimate is $\hat{\phi}_i = \sum_{s=1}^n m_s \hat{\phi}_{i,s}$ where $\hat{\phi}_{i,k} = \frac{1}{T} \sum_{j=1}^T \left( U(\mathcal{S}^k(\pi_j) \cup i) - U(\mathcal{S}^k(\pi_j)) \right)$ and $\mathcal{S}^k(\pi_j)$ is the subset that contains the first $k-1$ players in $\pi_j$.

**The SHAP-IQ Estimator (Fumagalli et al. 2024)**   Recall that its underlying formula is

$$\phi_i = p_n \cdot (U([n]) - U(\emptyset)) + 2H \cdot \mathbb{E}_{\emptyset \subsetneq S \subsetneq [n]}[((n-s)m_s \mathbb{1}_{i \in S} - sm_{s+1} \mathbb{1}_{i \notin S}) \cdot (U(S) - U(\emptyset))]$$

where $m_s = \binom{n-1}{s-1} p_s$, $H = \sum_{j=1}^{n-1} \frac{1}{j}$, and $P(S) \propto \binom{n-2}{s-1}^{-1}$. Therefore, a sequence of subsets $\{S_j\}_{j=1}^T$ where $\emptyset \subsetneq S_j \subsetneq [n]$ is sampled using $P(S) \propto \binom{n-2}{s-1}^{-1}$, and the $i$-th estimate is

$$\hat{\phi}_i = p_n \cdot (U([n]) - U(\emptyset)) + \frac{2H}{T} \sum_{j=1}^T (U(S_j) - U(\emptyset)) \cdot \left( (n-s)m_s \mathbb{1}_{i \in S_j} - sm_{s+1} \mathbb{1}_{i \notin S_j} \right).$$

