# OpenReview forum: "One Sample Fits All: Approximating All Probabilistic Values Simultaneously and Efficiently"
_NeurIPS.cc/2024/Conference — NeurIPS 2024 poster_

### Official Review · Reviewer_YTCS · 2024-07-05

**Soundness:** 3
**Presentation:** 3
**Contribution:** 2
**Rating:** 6
**Confidence:** 2

**Summary:**

This paper studies efficient estimation of probabilistic values. It proposes an algorithm which can approximate any probabilistic value with average convergence $O(n \log n)$. It also proposes an improved algorithm for specific cases.

**Strengths:**

This paper provides a solid contribution over previous work.

The paper is well written and explains the main ideas clearly.

**Weaknesses:**

If I’m understanding correctly, the algorithm approximates “any” instead of “all” probabilistic values, I.e. it’s a generic algorithm which can approximate any probabilistic value, but it can’t approximate all values simultaneously (one needs to rerun the algorithm for different values).

How valuable is the average convergence rate? It’s likely that the easy ones contribute more, and the average rate for the cases of interest is worse.

**Questions:**

See weaknesses.

**Limitations:**

See weaknesses.

---

> ### Author Rebuttal · Authors · 2024-08-06
>
> We are grateful for your review! Here is our response to your concerns.
>
> **Q: If I’m understanding correctly, the algorithm approximates “any” instead of “all” probabilistic values, i.e. it’s a generic algorithm which can approximate any probabilistic value, but it can’t approximate all values simultaneously (one needs to rerun the algorithm for different values).**
>
> A: We'll make it clearer on this part in the revision. As presented in line 12 of Algorithm 1, the coefficients $ \{ m_{s} \} $ (which vary for different probabilistic values) do not appear in the approximation procedure, i.e., Lines 1-11, meaning that the estimates $ \\{ \hat{\phi}\_{i,k^+}\^{+}, \hat{\phi}\_{i,k^-}\^{-} \\} $ can be translated to any probabilistic value. Therefore, the proposed algorithm is capable of sampling subsets **once** and then obtain the approximations of **all** probabilistic values. There is no need to rerun the algorithm (line 1-11) for different values; only the last aggregation step (which is trivial) needs to be rerun (line 12).
>
> **Q: How valuable is the average convergence rate? It’s likely that the easy ones contribute more, and the average rate for the cases of interest is worse.**
>
> A: We agree that not all probabilistic values may be of interest. Nevertheless, as proved in Proposition 2, our OFA-A estimator still achieves the convergence rate $ O(n\log n) $ **simultaneously** for all Beta Shapley values of interest. To our knowledge, the Beta Shapley values are not easy to estimate and they have important applications in practice. For example, the previous best theoretical convergence rate $ O(n(\log n)^{2}) $ for Beta$ (1,1) $ (i.e., the well-known Shapley value) is achieved by the group testing estimator, while the rate $ O(n(\log n)^{3}) $ for Beta$ (\alpha,\beta) $ with $ (\alpha=1, \beta>1) $ or $ (\alpha>1, \beta=1) $ is instead achieved by the GELS estimator. Therefore, the significance of OFA-A is twofold: i) it improves the theoretical convergence rate for some Beta Shapley values and ii) it achieves the convergence rate $ O(n \log n) $ **simultaneously** for all Beta Shapley values of interest. These results indicate that the average rate in Proposition 1 is not merely contributed by ''easy'' probabilistic values.

---

> > ### Comment · Reviewer_YTCS · 2024-08-08
> >
> > Thank you for the clarification! I will maintain my score.

---

### Official Review · Reviewer_k6r4 · 2024-07-08

**Soundness:** 3
**Presentation:** 2
**Contribution:** 3
**Rating:** 6
**Confidence:** 1

**Summary:**

The paper discusses a novel framework for efficiently approximating probabilistic values, such as Beta Shapley values and weighted Banzhaf values. These values are computationally expensive to calculate exactly, calling for approximation techniques.

Specifically, they propose the One-sample-Fits-All Framework (OFA), which adheres to the principle of maximum sample reuse and not increasing variance. Besides, two variants, OFA-A (and OFA-S) optimize for all probabilistic values on average (and specific probabilistic values), respectively.

Overall, they provide an efficient solution for approximating multiple probabilistic values simultaneously, with theoretical and practical implications.

**Strengths:**

1. The quick approximation of probabilistic values is important.

2. The writing flow is clear and easy to follow (especially Sec. 3).

3. The theoretical part seems reasonable.

**Weaknesses:**

1. The variances (and/or biases) of the estimations are not thoroughly examined; instead, they only consider whether the variances will increase based on the range/value of $m_s$.

2. The authors emphasize "the principle of maximum sample reuse" all the time, and treat it as a key principle. However, they do not provide a formal definition of this principle.

3. Similarly, a formal definition of "one-for-all estimators" is missing.

4. Line 95 mentions that an existing work already achieves $O(n/\epsilon^2 \log (n/\sigma))$, while the proposed OFA-A is $O(n \log (n))$. The significance of the contribution is unclear.

5. What does $q_s$ in line 159 mean?

**Questions:**

See the weaknesses part.

**Limitations:**

The limitations are not discussed explicitly in the main text.

---

> ### Author Rebuttal · Authors · 2024-08-06
>
> Thank you for your comments! This response is to address your concerns.
>
> **Q: The variances (and/or biases) of the estimations are not thoroughly examined; instead, they only consider whether the variances will increase based on the range/value of $ m_{s} $.**
>
> A: Please let us know if our understanding of your question is not correct, and we'd be happy to respond to any further questions.
>
> Our argument starts from the proof of the convergence rates of the mentioned estimators, where Hoeffding's inequality is always a key step. As a simple demonstration that applies to Eq. (1), let $ \\{ X_i \\}\_{i=1}\^{T} $ be $ T $ i.i.d. random variables such that $ X_{i} \in [a,b] $. (The non-identical case may be more complicated but the intuition remains the same.) Then, by Hoeffding's inequality, $ P(|c\cdot\overline{X} - c\cdot\mathbb{E}[X_{i}] |\geq \epsilon) \leq 2\exp\left( -\frac{2T\epsilon^{2}}{c^{2}(b-a)^{2}} \right) $ where $ \overline{X} = \frac{1}{T}\sum_{i=1}^{T}X_{i} $. Solving $ 2\exp\left( -\frac{2T\epsilon^{2}}{c^{2}(b-a)^{2}}\right) \leq \delta $, we have $ T \geq \frac{2c^{2}(b-a)^{2}}{\epsilon^{2}}\log\frac{2}{\delta} $, from which we observe that to improve the convergence rate (equivalently, to minimize $ T $) it is necessary to make $ c^{2} $ as small as possible. Meanwhile, $ \mathrm{Var}[c\cdot X_{i}] \leq \frac{c^{2}(b-a)^{2}}{4} $ and the equality could hold, indicating that $ c> 1 $ amplifies the worst variance. Intuitively, the above argument illustrates that for $ m_{s} > 1 $, the theoretical convergence rate may deteriorate due to amplified variance.
> Moreover, we have empirically verified that this indeed leads to slower empirical convergence in Figure 1. We'll make this part clearer in the revision.
>
>
> **Q: The authors emphasize "the principle of maximum sample reuse" all the time, and treat it as a key principle. However, they do not provide a formal definition of this principle.**
>
> A: We would like to elaborate on what is mentioned in lines 117-118 regarding the principle of maximum sample reuse. Precisely, any of the considered estimators iteratively samples a subset $ S \subseteq [n] $ and then uses $ U(S) $ to update **some** of the current estimates $ \\{ \hat{\phi}_i \\}\_{i\in [n]} $. An estimator is said to meet the principle of maximum sample reuse if every $ U(S) $ is employed in updating **all** the current estimates. We will formally define this principle in our revision.
>
>
> **Q: Similarly, a formal definition of "one-for-all estimators" is missing.**
>
> A: We'd like to add more details to our definition of one-for-all estimators. Previous works only considered the scenario of approximating **one** specific probabilistic value. As a result, the designed procedure may not be able to reuse the sampled subsets for any other probabilistic values (without amplifying variance). By contrast, a one-for-all estimator is able to sample subsets **once** and then obtain approximations for **all** probabilistic value. We will formally define one-for-all estimators in our revision. Thank you.
>
> **Q: Line 95 mentions that an existing work already achieves $ O(\frac{n}{\epsilon^{2}}\log\frac{n}{\delta}) $, while the proposed OFA-A is $ O(n\log n) $. The significance of the contribution is unclear.**
>
> A: Thank you for pointing out the misleading information contained in line 95. Precisely, Wang and Jia (2023) proposed an efficient estimator **only** for the Banzhaf value, and their convergence rate analysis is exclusive to the Banzhaf value. In contrast, OFA-A only needs to collect samples once which can then be used to approximate all probabilistic values (including Banzhaf). We have corrected this misprint.
>
> The significance of our theoretical results, e.g., includes:
>
> - As indicated by Proposition 2, the proposed OFA-A estimator achieves the convergence rate $ O(n\log n) $ **simultaneously** (i.e., sample subsets once to obtain all approximations) for all Beta Shapley values of interest. In contrast, the previous convergence analyses are limited to only **one** specific probabilistic value;
>
> - The well-known Shapley value belongs to the family of Beta Shapley values of interest and the previous best convergence rate $ O(n(\log n)^{2}) $ is achieved by the group testing estimator. By contrast, our proposed OFA-A achieves a slightly better rate $ O(n\log n) $.
>
> **Q: What does $ q_{s} $ in line 159 mean?**
>
> A: $ q_{s} $ used in Algorithm 1 is the probability of sampling a subset from $ \\{ T \subseteq [n] \mid |T| = s+1 \\} $. We have made it clearer in the revision.

---

> > ### Comment · Reviewer_k6r4 · 2024-08-09
> >
> > Thanks for the responses. I have raised my score from 5 to 6.

---

### Official Review · Reviewer_DfL8 · 2024-07-21

**Soundness:** 3
**Presentation:** 3
**Contribution:** 3
**Rating:** 6
**Confidence:** 1

**Summary:**

This paper presents a novel framework called One-Sample-Fits-All (OFA) for efficiently approximating probabilistic values used in feature attribution and data valuation, such as Beta Shapley values and weighted Banzhaf values. The framework maximizes sample reuse and avoids variance amplification, making it capable of approximating all probabilistic values simultaneously. The authors leverage the concept of
(ϵ,δ)-approximation to derive a formula that determines the convergence rate, which they use to optimize the sampling vector. The proposed framework includes two estimators: OFA-A, which is optimized for all probabilistic values on average, and OFA-S, which is fine-tuned for each specific probabilistic value. Empirical results demonstrate that OFA-A achieves the fastest known convergence rate for Beta Shapley values and competitive rates for weighted Banzhaf values.

**Strengths:**

1. The paper introduces a novel one-sample-fits-all framework that can approximate a wide range of probabilistic values efficiently, addressing a significant gap in the literature. The use of (ϵ,δ)-approximation to derive convergence rates and optimize the sampling vector adds strong theoretical backing to the proposed method.

2. The OFA-A estimator achieves the best known time complexity for certain probabilistic values, indicating its computational efficiency.

**Weaknesses:**

1. The empirical results are demonstrated on a classification task using LeNet and small datasets like MNIST.

2. The paper does not discuss much about the implementation details - the theoretical concepts and optimization processes might be complex for practitioners without a strong background in cooperative game theory and advanced statistical methods.

**Questions:**

The paper does provide an extensive theoretical proof regarding scalability of the pipeline, but there is limited explorations in terms of  emperical results for scalable real life datasets and problems. It would be nice to see some comments on that. While the scalability is theoretically supported, further empirical validation on larger datasets would strengthen the claim.

**Limitations:**

Yes the authors have indicated the weakness of their proposed method

---

> ### Author Rebuttal · Authors · 2024-08-06
>
> We appreciate your review and address your concerns below.
>
> **Q: The paper does not discuss much about the implementation details.**
>
> A: We agree that implementation details are important. We included the pseudocode in the submission and we will release our pytorch code after this work is made public.
>
> **Q: The paper does provide an extensive theoretical proof regarding scalability of the pipeline, but there is limited explorations in terms of empirical results for scalable real life datasets and problems.**
>
> A: In our experiments, we restricted the size of datasets for the sake of computing the **exact** groundtruth values, to which we can compare each baseline. Without the ground-truth values, we wouldn't be able to plot the curves in Figure 1.
> In practice where $ n $ is large, different baselines will return different approximations and it would be impossible to claim which is better **in terms of convergence**, but these approximations can always be used for some downstream applications, such as noisy label detection.

---

> > ### Comment · Reviewer_DfL8 · 2024-08-08
> >
> > Thank you for the insightful discussion!

---

### Official Review · Reviewer_2hZa · 2024-07-26

**Soundness:** 3
**Presentation:** 3
**Contribution:** 3
**Rating:** 7
**Confidence:** 3

**Summary:**

The paper studies efficient estimators for probabilistic values, with applications in data valuation and feature attribution. Since the computation of probabilistic values requires an exponential number of utility function evaluations, efficient estimation of probabilistic values is necessary. Existing approaches for this either contain amplifying scalers that degrade the convergence rate or fail to perform maximum sample re-use (i.e. utility evaluations are not used to update the probabilistic value estimates for all players). The paper proposes an algorithm that solves both the above challenges at once, thus providing a one-for-all estimator that achieves superior time complexity on average. The theoretical results are supported by several experimental results that validate improvements in speed of convergence as compared to the previous methods.

**Strengths:**

The paper provides a simple algorithm that updates the probabilistic values of all n players at once, thus improving our ability to compute multiple probabilistic values and allowing us to evaluate the best one for downstream applications. Since the exponential number of utility evaluations hinders the ability to perform this evaluation, this paper could be an important step towards understanding how to improve the time complexity of such methods. The paper carefully compares existing methods under the axes of amplifying scalers and maximum sample reuse, providing an intuitive goal for improved estimation algorithms. The experimental assessment supports the theory developed in the paper regarding (\epsilon, \delta) convergence rates.

**Weaknesses:**

The proposed algorithm depends on obtaining a good sampling vector q. Although the paper addresses this in sections 4.1 and 4.2, it would be helpful to discuss this further (e.g., challenges in solving the minimization problem in 4.1, alternatives to the faster generic estimator in 4.2, etc).

The experimental results vary the size of the dataset up to 256. In real-world data valuation tasks with larger sample sizes, would it be harder to scale this method despite beating all the baselines?

It would be helpful to discuss the limitations of the proposed approach both in terms of theoretical analysis and applicability to tasks like data valuation (It does talk about the limitations under proposition 3, but it would be helpful to have a separate discussion on limitations)

**Questions:**

Please see the weaknesses

**Limitations:**

Although broader societal impacts may not apply to this paper, this work should have a section on the limitations of the proposed algorithm in a separate section in addition to the brief discussion in Proposition 3.

---

> ### Author Rebuttal · Authors · 2024-08-06
>
> Thank you for the thoughtful comments. We address your concerns below.
>
> **Q: Discuss further the sampling vector q.**
>
> A: Thank you for this suggestion. The optimization problems in Section 4.1 and 4.2 are solved in closed-form, see Proposition 1 and Eq. (4). These closed-form solutions can be computed in O(n) time.
>
> The two sampling vectors are complementary to each other: using $\mathbf{q}^{\mathrm{OFA-A}}$ we can collect samples and (re)use them for any probabilistic value, while $\mathbf{q}^{\mathrm{OFA-S}}$ achieves better performance but depends on the underlying probabilistic value.
>
> We have added the above discussion and limitation to our revision.
>
> **Q: The experimental results vary the size of the dataset up to 256. In real-world data valuation tasks with larger sample sizes, would it be harder to scale this method despite beating all the baselines?**
>
> A: In our experiments, we restricted the size of datasets for the sake of computing the **exact** groundtruth values, to which we compare each baseline. In practice, the limiting factor (for any valuation method) is on computing the utility $U(S)$, which often involves training a (large) deep neural network on the subset $S$ of training data. To scale up, one can simply train the network with few active layers (from a pre-trained model). This heuristic is used in many existing valuation methods and applies equally well to our method.

---

> > ### Comment · Reviewer_2hZa · 2024-08-12
> >
> > Thank you for the clarifications. I am keeping my current rating.

---

### Decision · Program_Chairs · 2024-09-25

**Decision:**

Accept (poster)

**Comment:**

**Summary:** This paper proposes an estimation framework for all probabilistic values (such as Shapley values), where the approximation guarantees are given simultaneously across all values.

**Novelty and Clarity:** The perspective of simultaneously approximating all probabilistic values is novel and the topic of values is itself very timely. The paper is written and organized well, although certain expressions used (such as "maximum sample reuse" and “one-for-all estimators") are idiosyncratic and need to be defined and discussed, and certain connections to prior work need to be done more carefully, as corrected in the discussions.

**Significance:** One thing that is uncertain is whether simultaneous estimation is indeed important in downstream tasks. This is used as part of the motivation of the proposed perspective but is not elaborated on eventually, which diminished from the significance of the results. Another issue raised by reviewers is that the requisites for making the method work may not always be available (e.g., a good sampling vector). Also, empirical evaluation is hard to judge because, due to intractability, the experiments can’t quite scale to show that the method works in larger problems.

***

Not connecting the importance of estimating all values simultaneously to downstream tasks somewhat weakens the significance of the work. However, with a few pointers in this direction and ironing out some of the terminology and connections to prior work, this can become a solid paper on a timely and potentially impactful topic.